# Underground isoleucine biosynthesis pathways in *E. coli*

Charles AR Cotton[1], Iria Bernhardsgrütter[2], Hai He[1], Simon Burgener[2], Luca Schulz[2], Nicole Paczia[2], Beau Dronsella[1], Alexander Erban[1], Stepan Toman[1], Marian Dempfle[1], Alberto De Maria[1], Joachim Kopka[1], Steffen N Lindner[1], Tobias J Erb[2,3], Arren Bar-Even[1]*

[1]Max Planck Institute of Molecular Plant Physiology, Potsdam, Germany; [2]Max Planck Institute for Terrestrial Microbiology, Marburg, Germany; [3]LOEWE Research Center for Synthetic Microbiology (SYNMIKRO), Marburg, Germany

**Abstract** The promiscuous activities of enzymes provide fertile ground for the evolution of new metabolic pathways. Here, we systematically explore the ability of *E. coli* to harness underground metabolism to compensate for the deletion of an essential biosynthetic pathway. By deleting all threonine deaminases, we generated a strain in which isoleucine biosynthesis was interrupted at the level of 2-ketobutyrate. Incubation of this strain under aerobic conditions resulted in the emergence of a novel 2-ketobutyrate biosynthesis pathway based upon the promiscuous cleavage of *O*-succinyl-L-homoserine by cystathionine γ-synthase (MetB). Under anaerobic conditions, pyruvate formate-lyase enabled 2-ketobutyrate biosynthesis from propionyl-CoA and formate. Surprisingly, we found this anaerobic route to provide a substantial fraction of isoleucine in a wild-type strain when propionate is available in the medium. This study demonstrates the selective advantage underground metabolism offers, providing metabolic redundancy and flexibility which allow for the best use of environmental carbon sources.

*For correspondence:
Bar-Even@mpimp-golm.mpg.de

Competing interests: The authors declare that no competing interests exist.

## Introduction

The patchwork model suggests that novel metabolic pathways emerge from the promiscuous activities of enzymes participating in diverse metabolic processes (*Lazcano and Miller, 1999*; *Jensen, 1976*; *Khersonsky and Tawfik, 2010*; *Noda-Garcia et al., 2018*). According to this, underground metabolism – the network of metabolic conversions which are catalyzed as side reactions of enzymes that have evolved to support other activities (*D'Ari and Casadesús, 1998*) – provides fertile ground for the evolution of new pathways. A computational analysis suggested that about half of all underground reactions generate metabolites that already exist in the endogenous metabolic network, thus enabling the emergence of metabolic bypasses for the production of key cellular building blocks (*Notebaart et al., 2014*). For example, promiscuous activities of the arginine biosynthesis enzymes enabled proline production in an *E. coli* strain deleted in the canonical proline biosynthesis route (*Itikawa et al., 1968*); several pathways, each based on the promiscuous activities of different enzymes, have been shown to relieve pyridoxal phosphate auxotrophy in *E. coli* (*Kim et al., 2010*; *Oberhardt et al., 2016*); and adaptive evolution of *E. coli* harnessed enzyme promiscuity to enable growth on the non-natural feedstock 1,2-propanediol (*Lee and Palsson, 2010*).

The biosynthesis of isoleucine provides multiple examples of enzyme promiscuity and structural similarity to other pathways, suggesting underground metabolism as a likely origin (*Jensen, 1976*). Cellular production of isoleucine and valine are catalyzed by the same enzymes; in valine biosynthesis, pyruvate is self-condensed whereas in isoleucine biosynthesis, pyruvate is condensed with 2-ketobutyrate (2 KB). In most organisms, 2 KB is produced from threonine cleavage. Despite this, different biosynthetic routes, which mirror other metabolic pathways, are known to support the

production of 2 KB in specific lineages (*Figure 1*). For example, the citramalate route mirrors the TCA cycle, where pyruvate, rather than oxaloacetate, is condensed with acetyl-CoA, and 2 KB, rather than 2-ketoglutarate, is the pathway product (*Charon et al., 1974*). Similarly, in several anaerobic microorganisms, propionate is converted to 2 KB in a reductive pathway that mirrors acetate conversion to pyruvate via ligation with CoA and ferredoxin-dependent carboxylation (*Buchanan, 1969*; *Monticello et al., 1984*; *Eikmanns et al., 1983*). In plants, methionine γ-lyase, catalyzing a reaction similar to that of cystathionine γ-lyase, is a complementary source of cellular 2 KB (*Joshi and Jander, 2009*; *Joshi et al., 2010*).

Here, we tested whether the underground metabolism of *E. coli* could compensate for the deletion of the canonical 2 KB production route in isoleucine biosynthesis. Under aerobic conditions, we show that a mutation in a cysteine biosynthesis enzyme decreased its activity and lowered the steady-state concentration of this amino acid; this freed an intermediate of methionine biosynthesis, which usually reacts with cysteine, to be converted to 2 KB. Under anaerobic conditions, *E. coli* was able to use the enzyme pyruvate formate-lyase (PFL) to generate 2 KB from propionyl-CoA and formate. Surprisingly, we found this pathway to provide a substantial fraction of the cellular 2 KB also in a wild-type (WT) strain. This study thus demonstrates the inherent ability of microorganisms to effectively exploit underground metabolism to generate patchwork pathways.

## Results

### A latent aerobic isoleucine biosynthesis pathway

Biosynthesis of isoleucine in *E. coli* starts with the deamination of threonine to give 2 KB, which is then condensed with pyruvate to produce 2-aceto-2-hydroxybutanoate. This intermediate is subsequently isomerized, reduced, dehydrated, and aminated to generate isoleucine. To generate an isoleucine auxotrophy, we constructed a strain in which the two genes encoding for threonine deaminase were deleted (*ilvA* and *tdcB*). When provided with isoleucine, this strain grew identically to the WT strain (dark red line in *Figure 2*). Without the addition of isoleucine, we observed no growth in the first 70 hr, but within 70–120 hr, different replicates started growing with a growth rate somewhat lower than that of the WT strain (purple lines in *Figure 2*).

One explanation of the observed growth without isoleucine is an increased activity of serine deaminases, which are known to accept threonine as a substrate (*Cicchillo et al., 2004*). We, therefore, deleted all the genes encoding for serine deaminase (*sdaA*, *sdaB*, and *tdcG*). We termed the

**Figure 1.** Metabolic routes for 2-ketobutyrate biosynthesis. A wide variety of pathways support the production of 2-ketobutyrate, an essential intermediate in the production of isoleucine. Outlined are five known, physiologically relevant 2-ketobutyrate synthesis pathways and two additional pathways we identified in this paper (marked in blue). The purple font indicates the key enzymes of the novel 2-ketobutyrate production pathways in vivo. Note, the pathways are represented in shorthand between identifiable intermediates and do not show each metabolic step.

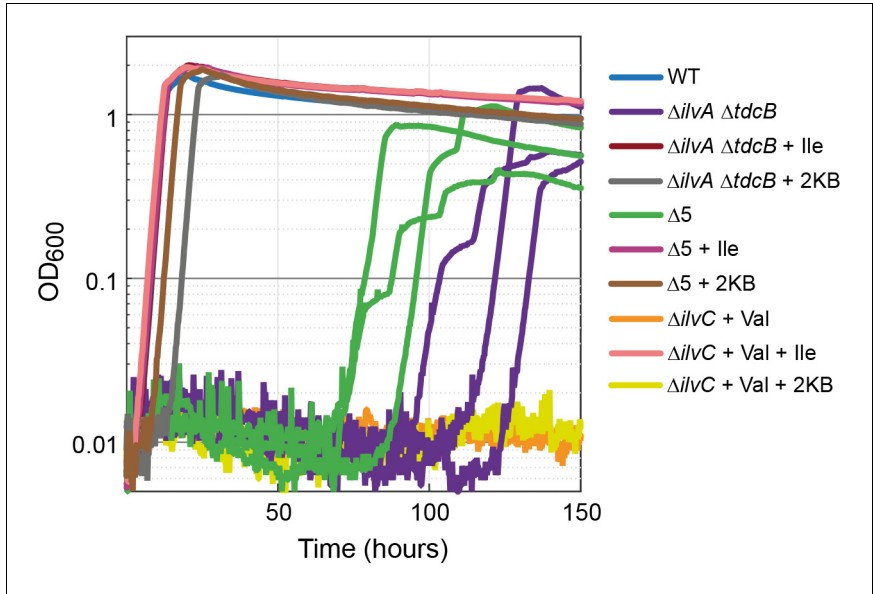

**Figure 2.** Strains lacking threonine deaminases adapt to generate isoleucine under aerobic conditions. Strains deleted in threonine deaminases (ΔilvA ΔtdcB) or threonine deaminases and serine deaminases (Δ5, i.e. ΔilvA ΔtdcB ΔsdaA ΔsdaB ΔtdcG) grew on minimal medium with 10 mM glucose after approx. 70 hr. Strains deleted in ketol-acid reductoisomerase (ΔilvC, an enzyme downstream of 2 KB in isoleucine biosynthesis) could not grow (valine supplied to relieve a second auxotrophy), indicating a metabolic leak at the level of 2 KB. All experiments were performed in technical triplicates in a 96-well plate reader. For the ΔilvA ΔtdcB and Δ5 strains, cultivated without isoleucine (purple and green lines), replicates are shown individually. In all other cases, replicates showed an identical growth profile (±5%) and hence are represented by a single curve. Experiments were repeated on three separate occasions for all growth experiments shown. Isoleucine, 2-ketobutyrate, and valine were added at 2 mM where indicated.

resulting strain Δ5 (ΔilvA ΔtdcB ΔsdaA ΔsdaB ΔtdcG). However, strain Δ5 displayed similar behavior to the ΔilvA ΔtdcB strain, that is, while growth without isoleucine was not observed in the first 70 hr, after 70 hr, the replicates started growing (green lines in *Figure 2*). This strongly suggests the emergence of a latent threonine-independent isoleucine biosynthesis pathway.

To check whether this route still depends on the generation of 2 KB or rather bypasses this metabolic intermediate altogether, we constructed a strain deleted in the gene *ilvC*. This gene encodes a ketol-acid reductoisomerase that operates downstream of 2 KB in the isoleucine biosynthesis pathway. While cultivation of the ΔilvC strain with valine and isoleucine enabled growth similar to that of the WT strain (pink line in *Figure 2*; valine is required as IlvC participates also in valine biosynthesis), growth without isoleucine was not observed even after 150 hr (orange line in *Figure 2*). As a further confirmation, we observed that addition of 2 KB to the cultivation medium rescued the growth of the ΔilvA ΔtdcB strain and the Δ5 strain but not the ΔilvC strain (*Figure 2*). This confirms that 2 KB still serves as a metabolic intermediate in the underground isoleucine biosynthesis pathway.

In some microorganisms, 2 KB is produced via the citramalate pathway (*Xu et al., 2004*; *Drevland et al., 2007*; *Hochuli et al., 1999*; *Risso et al., 2008*). This route is a catalytic parallel to the first half of the TCA cycle – from citrate synthase to isocitrate dehydrogenase – where pyruvate replaces oxaloacetate, reacting with acetyl-CoA to give citramalate, which is subsequently metabolized to 2 KB. We wondered whether the enzymes of the TCA cycle, or their isozymes, catalyze the reactions of the citramalate pathway in *E. coli*. To check if this were the case, we performed a carbon labeling assay, feeding the WT strain, the ΔilvA ΔtdcB strain, and the Δ5 strain with either glucose-1-$^{13}$C or glucose-3-$^{13}$C. The expected labeling pattern of isoleucine should completely change depending on the biosynthesis route of 2 KB (*Figure 3A*). In the WT strain, isoleucine is expected to be roughly half labeled upon feeding with glucose-3-$^{13}$C and follow a 1:2:1 pattern (unlabeled: once labeled: double labeled) with glucose-1-$^{13}$C. By contrast, if 2 KB is produced via the citramalate pathway, then isoleucine would be completely unlabeled when feeding with glucose-3-$^{13}$C and

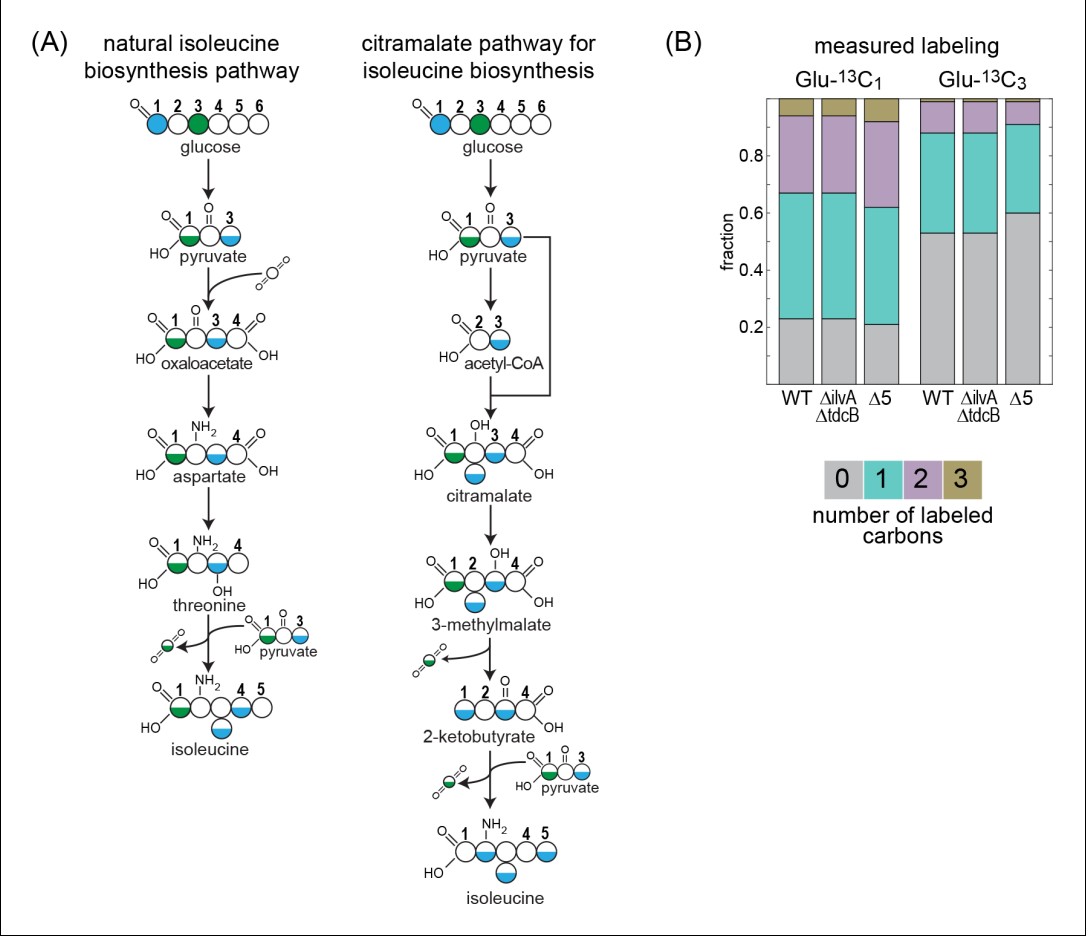

**Figure 3.** $^{13}$C-labeling of isoleucine rules out the citramalate pathway. Labeling of proteinogenic amino acids was analyzed after the growth of *E. coli* on labeled glucose-1-$^{13}$C or glucose-3-$^{13}$C as a sole carbon source. (**A**) Schematic representation of predicted labeling of isoleucine with the natural isoleucine biosynthesis pathway of *E. coli* (left) or with the citramalate pathway (right). Blue coloring highlights predicted labeling when grown on glucose-1-$^{13}$C and green coloring when grown on glucose-3-$^{13}$C. (**B**) Observed labeling pattern. There is no difference in the labeling pattern between the mutant strains and the WT strain, ruling out the citramalate pathway for isoleucine biosynthesis. Δ5 represents Δ*ilvA* Δ*tdcB* Δ*sdaA* Δ*sdaB* Δ*tdcG*. Labeling experiments were repeated on at least two separate occasions.

The online version of this article includes the following source data for figure 3:

**Source data 1.** Carbon labeling experiment.

would roughly follow a 1:2:2:1 pattern (unlabeled: once labeled: double labeled: triple labeled) with glucose-1-$^{13}$C. The labeling pattern of isoleucine in the Δ*ilvA* Δ*tdcB* and the Δ5 strains is almost identical to that of the WT strain (*Figure 3B*, the small deviations from the expected labeling can be attributed to the ambient abundance of $^{13}$C and the shuffling of labeled carbon by the activity of the pentose phosphate pathway and the TCA cycle). This effectively rules out the activity of the citramalate pathway and indicates that the biosynthesis of 2 KB in both gene deletion strains is closely related to the natural production pathway.

## 2-Ketobutyrate biosynthesis from *O*-succinyl-L-homoserine

In several organisms, the biosynthesis of isoleucine is at least partially tied to that of the sulfur-containing amino acids cysteine and methionine. For example, in organisms in which cysteine biosynthesis depends on γ-elimination of cystathionine, 2 KB serves as a byproduct (*Steegborn et al., 1999*). Also, in plants, 2 KB can be produced from γ-elimination of methionine (*Joshi and Jander, 2009*; *Joshi et al., 2010*). To test whether the production of 2 KB in the Δ5 strain is related to methionine

biosynthesis, we deleted *metA*, encoding for homoserine *O*-succinyltransferase, which catalyzes the first committed step of the methionine biosynthesis pathway. The Δ5 Δ*metA* strain could grow with the addition of methionine and isoleucine (*Figure 4*, pink line), while no growth was observed without isoleucine (*Figure 4*, orange line). This indicates that the production of 2 KB in the Δ5 strain is indeed dependent on the methionine biosynthesis route.

At first glance, the biosynthesis of sulfur-containing amino acids in *E. coli* would not be expected to give rise to 2 KB. Specifically, in this bacterium, cystathionine does not undergo γ-elimination but rather β-elimination which generates homocysteine and pyruvate (*Figge, 2006*). Also, *E. coli* lacks a characterized annotated methionine γ-lyase. While an *E. coli* gene was previously suggested to encode for such an enzyme (*Manukhov et al., 2005*), the inability of the Δ5 Δ*metA* strain to grow when supplemented with methionine but not isoleucine confirms that *E. coli* lacks methionine γ-lyase activity.

We, therefore, searched the literature for reactions that could convert intermediates of the methionine biosynthesis pathway to 2 KB. We found multiple such candidate reactions from a range of organisms (*Figure 5*). First, rather than catalyzing the condensation of cysteine with *O*-succinyl-L-homoserine, cystathionine γ-synthase (MetB) can cleave the latter intermediate to succinate and 2 KB (reaction one in *Figure 5*; $k_{cat}$ = 7.7 s$^{-1}$ and $K_M$(*O*-succinyl-L-homoserine)=0.33 mM for *E. coli* enzyme [*Figge, 2006*; *Holbrook et al., 1990*]). MetB can also condense *O*-succinyl-L-homoserine and homocysteine to give homolanthionine (reaction two in *Figure 5*; $K_M$(homocysteine)=0.54 mM for *C. glutamicum* enzyme [*Krömer et al., 2006*]), which can be subsequently cleaved by cystathionine-β-lyase (MetC) to regenerate homocysteine and produce 2 KB (reaction three in *Figure 5*; e.g. $k_{cat} \approx$ 180 s$^{-1}$ and $K_M$(homolanthionine)=4.5 mM for *E. coli* enzyme [*Figge, 2006*; *Krömer et al., 2006*; *Alting et al., 1995*; *Dwivedi et al., 1982*]). In addition, MetC from various organisms can catalyze both the β-elimination and the γ-elimination of cystathionine, giving rise to both pyruvate and 2 KB (reaction four in *Figure 5*; e.g. $k_{cat}$(γ-elimination) $\approx$ 0.9 s$^{-1}$ and $K_M$(cystathionine)=0.20 mM for *Streptomyces phaeochromogenes* enzyme where β-elimination proceeds at $\approx$ 1/7 rate of γ-elimination [*Alting et al., 1995*; *Nagasawa et al., 1984*]). As MetC can also cleave cysteine to pyruvate

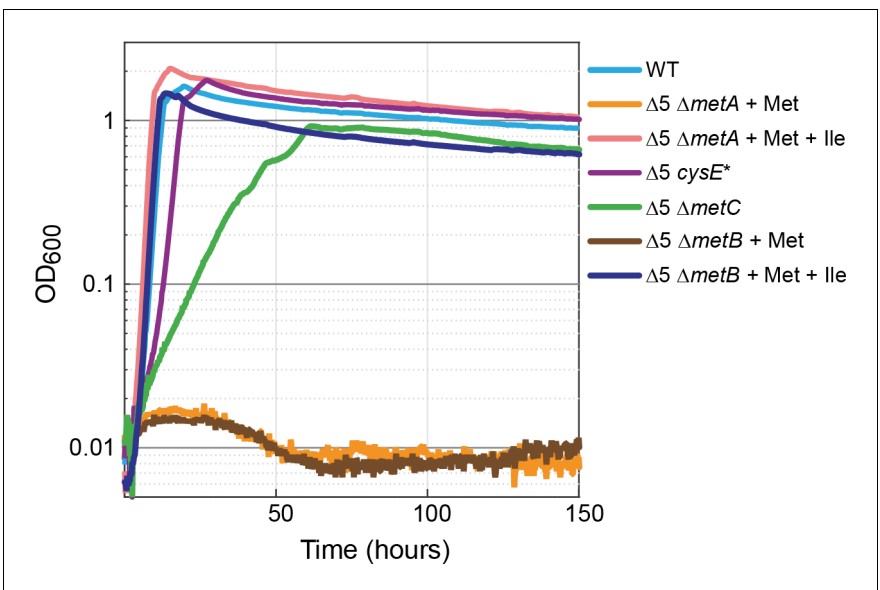

**Figure 4.** Methionine biosynthesis is implicated in promiscuous 2-ketobutyrate production. The Δ5 strain further deleted in homoserine *O*-succinyltransferase (Δ5 Δ*metA*) did not grow when supplemented with 2 mM methionine (orange line) but mutants of the cystathionine β-lyase (Δ5 Δ*metC*) grew rapidly and consistently on 10 mM glucose alone. A mutation in serine acetyltransferase (Δ5 *cysE**) resulted in near WT growth on glucose (purple line). Both isoleucine (Ile) and methionine (Met) were added at a concentration of 2 mM where indicated. All experiments were performed in replicates in a 96-well plate reader. Replicates showed an identical growth profile (±5%) and hence are represented by a single curve. Experiments were repeated at least three times for all growth experiments shown.

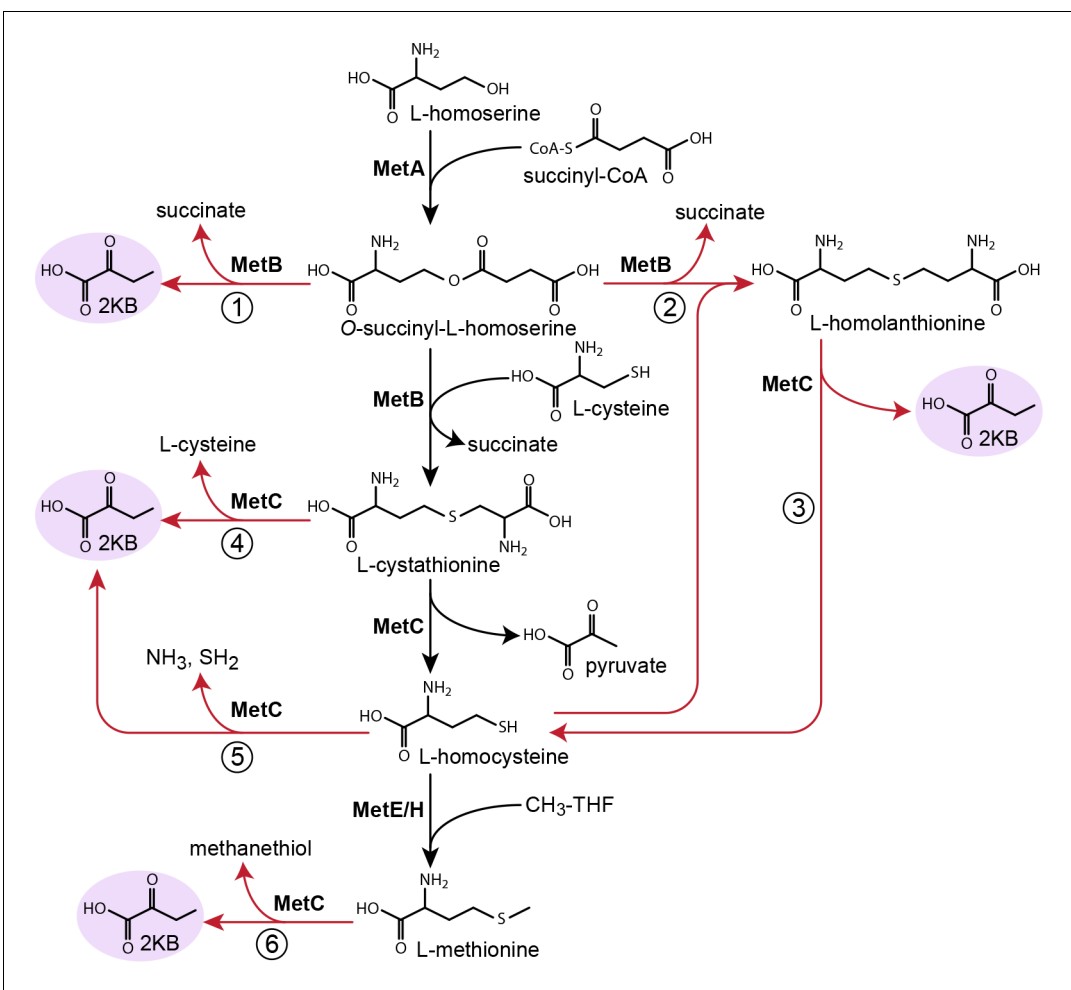

**Figure 5.** Possible routes to 2-ketobutyrate by methionine biosynthesis enzymes. Schematic representation of methionine biosynthesis from homoserine as operating in *E. coli* (black arrows). Promiscuous reactions of cystathionine γ-synthase (MetB) and cystathionine β-synthase (MetC) which have been suggested to produce 2-ketobutyrate in various organisms are shown as red arrows.

The online version of this article includes the following figure supplement(s) for figure 5:

**Figure supplement 1.** SDS-PAGE of all purified enzymes used in this study.

**Figure supplement 2.** MetB catalyzes *O*-succinyl-L-homoserine (OSHS) cleavage to generate 2-ketobutyrate (2 KB).

**Figure supplement 3.** Kinetic characterization of CysE variants.

($k_{cat} \approx$ 1.8 s$^{-1}$ for *E. coli* enzyme [*Dwivedi et al., 1982*; *Awano et al., 2003*; *Flint et al., 1996*]), it is possible that it could also cleave homocysteine to produce 2 KB (reaction five in *Figure 5*). Finally, some MetC variants can act as methionine γ-lyase, releasing 2 KB by directly cleaving methionine (reaction six in *Figure 5*; e.g. $k_{cat} \approx$ 0.01 s$^{-1}$ for *Lactococcus lactis* subsp. *cremoris* B78 enzyme [*Alting et al., 1995*; *Amarita et al., 2004*; *Irmler et al., 2008*]).

To identify which of the possible biosynthesis routes is responsible for the production of 2 KB in the Δ5 strain, we performed several enzymatic experiments (Materials and methods and *Figure 5—figure supplement 1*). We found that MetC from *E. coli* did not catalyze the degradation of cystathionine, homocysteine, or methionine to 2 KB (ruling out reactions 4, 5, and 6 in *Figure 5*). To further confirm that MetC is not involved in 2 KB production, we deleted its encoding gene. The Δ5 Δ*metC* strain could grow even without the addition of methionine (which is in line with the Δ*metC* strain in the Keio collection [*Baba et al., 2006*]), presumably due to the existence of multiple PLP-dependent enzymes that can catalyze the MetC reaction, for example MalY, Alr, and FimE

(*Zdych et al., 1995*; *Patrick et al., 2007*). More importantly, this strain could directly grow without isoleucine and without the need for adaptation (green line in *Figure 4*).

Supporting previous studies (*Figge, 2006*; *Holbrook et al., 1990*), we found MetB to catalyze the cleavage of *O*-succinyl-L-homoserine to 2 KB (reaction one in *Figure 5*) with $k_{cat}$ = 9.3 ± 0.4 s$^{-1}$, $K_M$(*O*-succinyl-L-homoserine)=0.60 ± 0.08 mM, and thus $k_{cat}/K_M$(*O*-succinyl-L-homoserine)=16 ± 2 mM$^{-1}$ s$^{-1}$ (*Figure 5—figure supplement 2A*). Yet, it was previously reported that the formation of 2 KB is suppressed in the presence of cysteine (*Holbrook et al., 1990*). We, therefore, characterized the cleavage reaction in the presence of either cysteine or homocysteine (as an alternative substrate), at physiological concentrations of ~0.3 mM (*Wheldrake, 1967*; *Bennett et al., 2009*; *Guo et al., 2013*) and at artificially high concentrations of 3–6 mM. We found that formation of 2 KB was indeed suppressed at physiological concentrations of cysteine, while homocysteine suppressed 2 KB formation only at high concentrations (*Figure 5—figure supplement 2B–E*). Nonetheless, as the deletion of *metB* in the Δ5 strain completely abolishes growth without isoleucine (brown line in *Figure 4*), it seems very likely that MetB is involved in the production of 2 KB.

## Disruption of MetC or a mutation in serine acetyltransferase enable steady 2-ketobutyrate production from *O*-succinyl-L-homoserine

Next, we aimed to understand the genetic basis underlying 2 KB biosynthesis in the Δ*ilvA* Δ*tdcB* and Δ5 strains. We sequenced the genomes of several strains isolated from the cultures growing without isoleucine. We found that in most of the sequenced strains (11 strains out of the 16 sequenced) *metC* was either deleted or mutated (*Supplementary file 1*). This is in line with the findings reported above, that is, Δ5 Δ*metC* strain is able to directly grow without isoleucine and without lag time (green line in *Figure 4*). It therefore seems that a lower metabolic flux toward methionine biosynthesis, as expected by the deletion of *metC*, enhances the side reactivity of the pathway enzymes and results in a higher conversion rate of *O*-succinyl-L-homoserine to 2 KB. Indeed, we found that, as compared to the WT strain and the Δ5 strain, in the Δ5 Δ*metC* strain the concentration of methionine was ≈3-fold lower (*Figure 6A*), while the concentration of *O*-succinyl-L-homoserine was ≈3-fold higher (*Figure 6A*). The side reactivity of MetB toward *O*-succinyl-L-homoserine cleavage therefore seems to be enhanced by the high concentration of this metabolite in the Δ5 Δ*metC* strain.

Another mutated strain harbored a single mutation in the gene coding for serine acetyltransferase (CysE): Ala33Thr (*Supplementary file 1*). We used Multiplex Automated Genomic Engineering (MAGE [*Wang et al., 2009*]) to introduce this mutation into the Δ5 strain. The resulting strain (Δ5 *cysE*\*) grew immediately (with no lag time) on a medium without isoleucine and with a growth rate identical to the WT strain (purple lines in *Figure 4*; note that the growth of the Δ5 *cysE*\* strain was considerably faster than that of the Δ5 Δ*metC* strain). Therefore, it seems that the mutation in *cysE* enabled a steady, efficient production of 2 KB.

CysE catalyzes the first committed step in the biosynthesis of cysteine (*Denk and Bock, 1987*). To understand the effect of the Ala33Thr mutation on enzyme activity we measured the kinetics of the purified enzyme (Materials and methods and *Figure 5—figure supplements 1* and *3*). Despite the mutation occurring far from the active site, we found that the apparent $k_{cat}$ of the enzyme decreased 2-fold, from 350 ± 30 s$^{-1}$ to 170 ± 30 s$^{-1}$, while the $K_M$ for acetyl-CoA increased by more than 8-fold, from 0.6 ± 0.2 mM to 5.0 ± 2.0 mM (interestingly, $K_M$ for serine changed only slightly from 0.8 ± 0.3 mM in the WT to 0.5 ± 0.2 mM in the mutant). As the concentration of acetyl-CoA in *E. coli* lies in the range of 0.6–0.75 mM (*Bennett et al., 2009*), the increase in $K_M$ for acetyl-CoA directly affects the reaction rate. Overall, the Ala33Thr mutation is expected to decrease the rate of the CysE reaction by more than 17-fold under physiological conditions.

The decreased activity of CysE halved the intracellular cysteine concentration relative to that of the WT strain and the Δ5 strain (*Figure 7*). This deprived MetB of its co-substrate, thus lowering the methionine biosynthesis flux and resulting in more than a 2-fold reduction in methionine concentration (*Figure 6A*). The lower availability of methionine led to overexpression of genes participating in its biosynthesis, increasing the transcript level of *metA* and *metB* more than 3-fold relative to the WT strain and the Δ5 strain (*Figure 6B*; interestingly, such increase in transcript level was not observed in the Δ5 Δ*metC* strain). This higher expression, as well as the partial lifting of MetA inhibition by methionine (*Born and Blanchard, 1999*), led to a dramatic 30- to 60-fold increase in the concentration of *O*-succinyl-L-homoserine (*Figure 6A*). Overall, the higher concentrations of both MetB and *O*-succinyl-L-homoserine, coupled with the lower concentration of cysteine, resulted in the

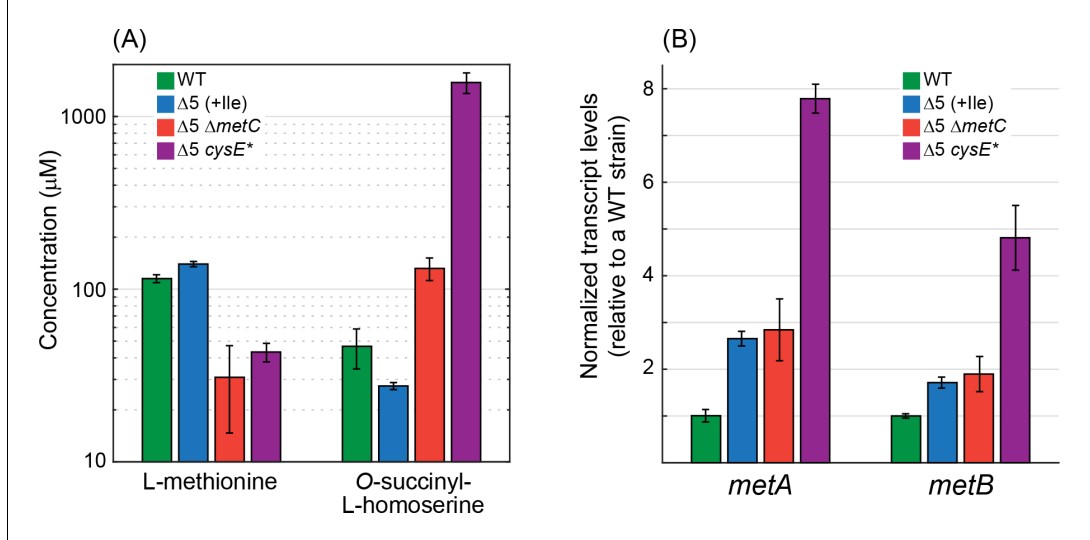

**Figure 6.** Deletion of *metC* or mutation in *cysE* substantially affects metabolite concentrations and gene expression. (A) Quantitative determination of the concentrations of methionine and *O*-succinyl-L-homoserine as performed using an HRES-LC-MS. The concentration of methionine in the Δ5 Δ*metC* and Δ5 *cysE*\* strains was 2- to 3-fold lower than in the WT strain and the Δ5 strain (p-value <0.05, rank sum test). Isoleucine was added only to the Δ5 strain (as it is required for its growth). Relative to the WT strain and the Δ5 strain, the concentration of *O*-succinyl-L-homoserine was 2- to 3-fold higher in the Δ5 Δ*metC* strain (p-value <0.05, rank sum test) and 30- to 60-fold higher in the Δ5 Δ*cysE*\* strain (p-value <0.05, rank sum test). Error bars correspond to standard deviations. (B) Quantitative determination of transcript levels of *metA* and *metB* as measured by reverse transcriptase quantitative PCR. The transcript levels of *metA* and *metB* were more than 3-fold higher in the Δ5 *cysE*\* strain than in the WT strain and the Δ5 strain (p-value <0.05, rank sum test). Error bars correspond to standard deviations.

The online version of this article includes the following source data for figure 6:

**Source data 1.** Measured concentrations of methionine and *O*-succinyl-L-homoserine.
**Source data 2.** Measured transcript levels of *metA* and *metB*.

diversion of more *O*-succinyl-L-homoserine toward cleavage and 2 KB production. Interpreted so, the mutation in CysE – an enzyme that is not directly involved in 2 KB biosynthesis – enhances a previously negligible underground reaction for 2 KB production, thus awakening a latent isoleucine biosynthesis route (*Figure 7*).

## Anaerobic 2-ketobutyrate biosynthesis from a reversible 2-ketobutyrate formate-lyase activity

Next, we aimed to explore underground isoleucine biosynthesis routes under anaerobic conditions (*Figure 8*). The evolved Δ*ilvA* Δ*tdcB cysE*\* strain and the constructed Δ5 *cysE*\* strain grew anaerobically without isoleucine (*Figure 8—figure supplement 1*). However, we did not observe the reemergence of the *metC* deletion, the *cysE*\* mutation, or any other mutation: the Δ5 strain failed to grow without the addition of isoleucine, even after 120 hr (black lines in *Figure 8A*). We wondered whether we could enable 2 KB production by the addition of small metabolites that *E. coli* might encounter in its natural habitat. We focused on propionate, a short-chain fatty acid abundant in the mammalian intestine (*McNeil et al., 1978*), which is known to be naturally activated to propionyl-CoA (*Hesslinger et al., 1998*; *Liu et al., 2014*). As we had previously found that PFL (encoded by *pflB*) supported the in vivo condensation of acetyl-CoA and formate to produce pyruvate (*Zelcbuch et al., 2016*), we hypothesized that this enzyme could also catalyze a 2 KB formate-lyase (KBFL) reaction, that is, accepting propionyl-CoA to produce 2 KB. We indeed found that a purified PFL catalyzes the condensation of propionyl-CoA and formate to give 2 KB with a specific activity of 0.6–1.1 μmol/min/mg, $k_{cat}$ of 0.9–1.7 s$^{-1}$, $K_M$(propionyl-CoA)=0.83 ± 0.6 mM and $K_M$(formate) =69 ± 11 mM (*Table 1*, Materials and methods, *Figure 5—figure supplement 1*, and *Figure 8—figure supplements 2* and *3*).

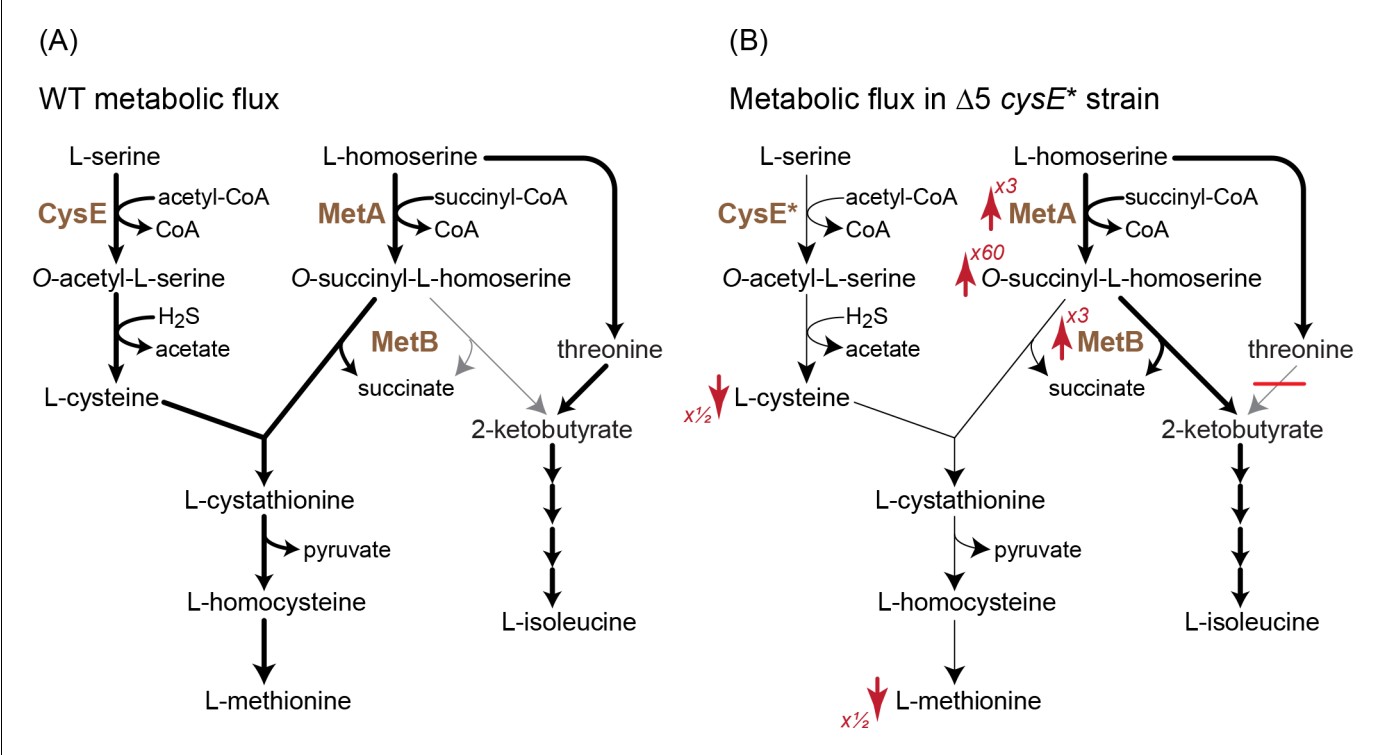

**Figure 7.** A mutation in serine acetyltransferase (*cysE**) enables 2-ketobutyrate production by MetB. (**A**) A schematic representation of endogenous cysteine, methionine, and isoleucine biosynthesis in *E. coli*. Thickness of arrows indicates flux intensity. (**B**) Schematic representation of cysteine, methionine, and isoleucine biosynthesis when threonine deaminases are deleted and with the *cysE** mutation. Thickness of the arrows indicates higher flux via *O*-succinyl-L-homoserine cleavage to 2-ketobutyrate. As indicated by the red arrows, the transcript levels of *metA* and *metB* as well the concentration of *O*-succinyl-L-homoserine increase, while the concentrations of cysteine and methionine decrease. Numbers next to the arrows correspond to the factor by which the metabolite concentration or transcript level increased or decreased (as derived from *Figure 6*). Cysteine concentrations in the Δ5 Δ*cysE** strain were ≈50% of that in WT and Δ5 strains.

The online version of this article includes the following source data for figure 7:

**Source data 1.** Relative concentrations of cysteine.

We found that the Δ5 strain could grow anaerobically when propionate and formate were added to the medium instead of isoleucine (red line in *Figure 8A*). As formate is naturally produced by the PFL-dependent cleavage of pyruvate, we reasoned that the addition of this compound to the medium might be redundant. Indeed, we found that propionate alone, when added to the medium, supported the growth of the Δ5 strain, albeit with longer lag time and lower growth rate than with the addition of formate (purple line in *Figure 8A*, all replicates displayed an identical growth phenotype, thus ruling out genetic mutations). To confirm that propionate is assimilated via the activity of PFL, rather than via a reductive carboxylation route (*Buchanan, 1969*; *Monticello et al., 1984*; *Eikmanns et al., 1983*), we deleted *pflB*. When provided with isoleucine and acetate, the Δ5 Δ*pflB* strain was able to grow (green line in *Figure 8B*, acetate was added as the deletion of *pflB* disrupted the endogenous acetyl-CoA biosynthesis route under anaerobic conditions [*Hasona et al., 2004*]). However, replacement of isoleucine with propionate, with or without further addition of formate, resulted in no growth (red and purple lines in *Figure 8B*). This confirms that PFL is indeed responsible for 2 KB biosynthesis.

To provide a further validation that 2 KB is produced from propionate, we followed the $^{13}C$ labeling of isoleucine upon addition of propionate-1-$^{13}C$ to the growth medium under anaerobic conditions. Surprisingly, we found that even within a WT strain, the addition of labeled propionate resulted in 30% labeled isoleucine (*Figure 9*), suggesting that the KBFL route operates naturally if propionate is present in the medium. When formate was also added to the medium, the fraction of labeled isoleucine increased to almost 60% (*Figure 9*), making the KBFL-dependent pathway the

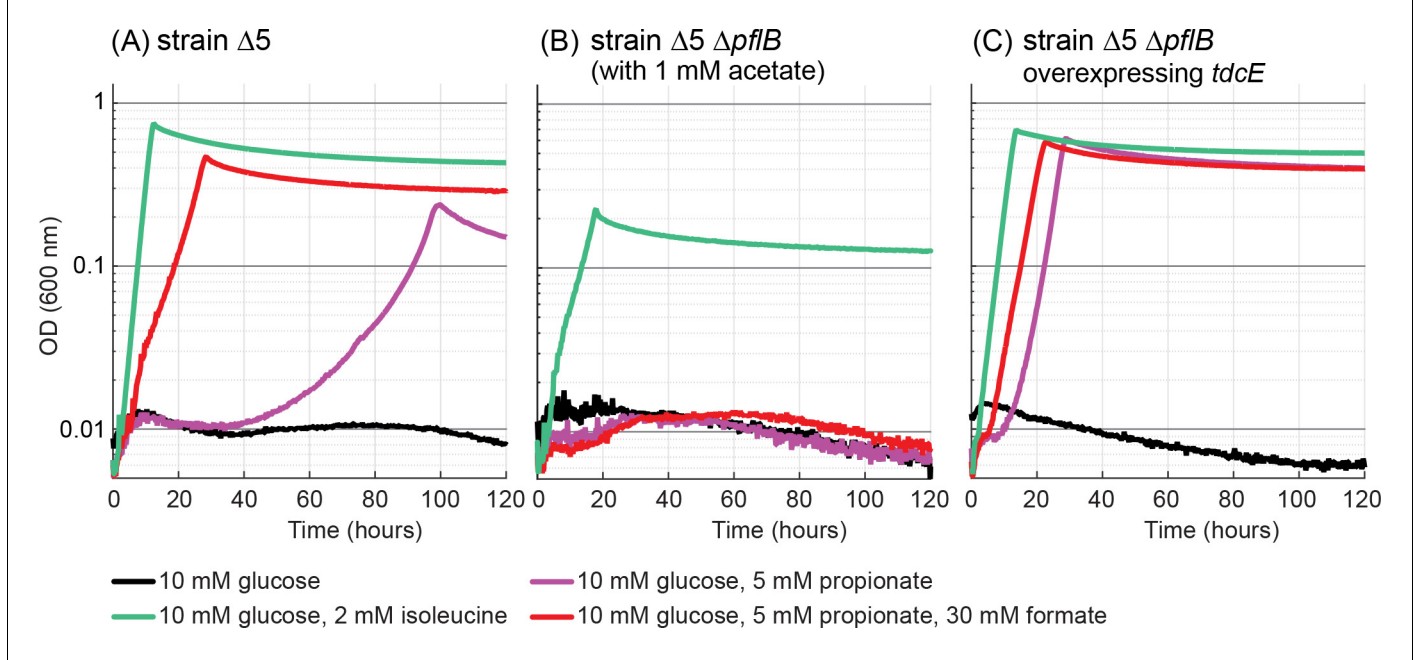

**Figure 8.** Propionate-dependent isoleucine production under anaerobic conditions. (**A**) Addition of 5 mM propionate enabled growth of the Δ5 strain under anaerobic conditions. Further addition of 30 mM formate improved growth substantially. (**B**) A Δ5 Δ*pflB* strain could not use propionate as precursor of isoleucine. 1 mM of acetate was added for the cultivation of this strain, as the deletion of *pflB* disrupted the endogenous acetyl-CoA biosynthesis. (**C**) Overexpression of *tdcE* in the Δ5 Δ*pflB* strain enabled growth with propionate as isoleucine precursor. All experiments were performed in replicates in a 96-well plate reader. Replicates showed an identical growth profile (±5%) and hence are represented by a single curve. Experiments were repeated three times for all growth experiments shown.

The online version of this article includes the following figure supplement(s) for figure 8:

**Figure supplement 1.** Growth via the MetB-dependent pathway under anaerobic conditions.

**Figure supplement 2.** In vitro activation and validation of PFL and TdcE activity (**A**) UV-Vis absorbance spectrum of 100 µM PFL-AE before and after reconstitution.

**Figure supplement 3.** Michaelis-Menten graphs of PFL and TdcE for propionyl-CoA and formate (see Materials and methods section for exact conditions).

dominant isoleucine biosynthesis route within a WT strain. In the Δ5 strain, in which the canonical isoleucine biosynthesis pathways are deleted, isoleucine was almost 80% labeled upon addition of formate and labeled propionate (*Figure 9*). Deletion of the *metA* increased this labeling to 90% (*Figure 9*), indicating that the MetB-dependent route operates anaerobically although with a low flux. As 10% of the isoleucine was not labeled even in the Δ5 Δ*metA* strain, it seems that there might be another undiscovered yet marginal route for isoleucine biosynthesis.

We wondered if the KBFL reaction is supported solely by PFL or whether TdcE, which was previously shown to catalyze the 2 KB formate-lyase reaction (*Sawers et al., 1998*), also contributed to the production of 2 KB. We found that purified TdcE is catalytically superior to PFL, generating 2 KB

**Table 1.** Kinetic parameters of PFL and TdcE for catalyzing the condensation of propionyl-CoA with formate to generate 2-ketobutyrate.

See Materials and methods for assay conditions and *Figure 8—figure supplements 2* and *3*. Note that the $k_{cat}$ values were calculated assuming that the enzymes are fully activated.

| Enzyme | Propionyl-CoA | | | | Formate | | |
|---|---|---|---|---|---|---|---|
| | $V_{max}$ (U/mg) | $k_{cat}$ (s$^{-1}$) | $K_M$ (µM) | $K_i$ (µM) | $V_{max}$ (U/mg) | $k_{cat}$ (s$^{-1}$) | $K_M$ (mM) |
| TdcE | 2.1 ± 0.1 | 3.2 ± 0.2 | 520 ± 81 | - | 1.8 ± 0.07 | 2.7 ± 0.1 | 20 ± 3 |
| PFL (PflB) | 1.1 ± 0.6 | 1.7 ± 0.9 | 830 ± 580 | 690 ± 500 | 0.59 ± 0.03 | 0.9 ± 0.05 | 69 ± 11 |

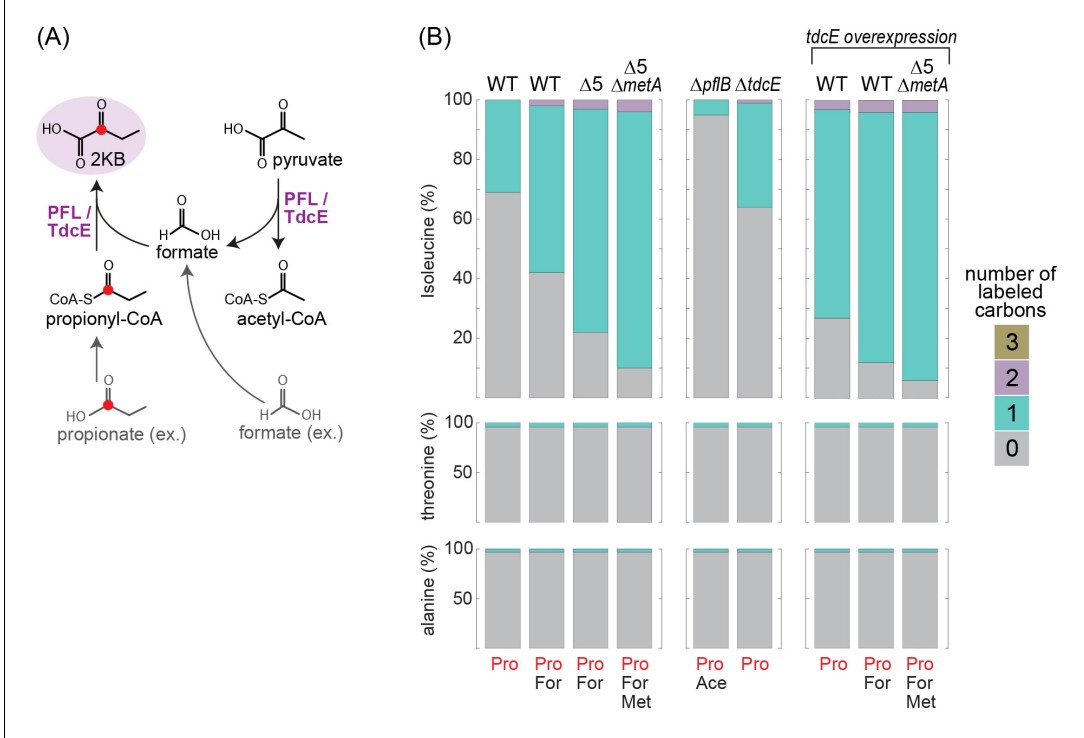

**Figure 9.** $^{13}$C-labeling of isoleucine reveals the activity of the KBFL route under anaerobic conditions. (A) Schematic representation of 2-ketobutyrate production via PFL/TdcE-catalyzed reaction. Compounds in gray are added to the medium (external) while those in black represent intracellular metabolites. The red circle indicates the labeled carbon. (B) 5 mM of propionate-1-$^{13}$C were added to the medium of different strains cultivated on 10 mM glucose. 10 mM acetate, 30 mM formate, and/or 2 mM methionine were added to the media of some strains, as indicated at the bottom of the figure. Alanine and threonine were not labeled in all cases, confirming that the labeling of isoleucine is derived from direct propionate conversion to 2 KB.

The online version of this article includes the following source data for figure 9:

**Source data 1.** Retention time, relative peak, and *m/z* for the carbon labeling experiment.

with a specific activity of 1.8–2.1 µmol/min/mg, $k_{cat}$ of 2.7–3.2 s$^{-1}$, $K_M$(propionyl-CoA)=0.52 ± 0.08 mM, and $K_M$(formate)=20 ± 3 mM (*Table 1*, Materials and methods, *Figure 5—figure supplement 1*, and *Figure 8—figure supplements 2* and *3*). Yet, while the deletion of *pflB* (in which case acetate was further added to the medium) abolished isoleucine labeling, deletion of *tdcE* resulted in identical labeling to that of the WT strain (*Figure 9*). This indicates that PFL is solely responsible for the production of 2 KB from propionate and formate. Still, it could be that TdcE does not contribute to 2 KB biosynthesis only because it is not expressed under the tested conditions. Hence, we explored the consequences of overexpressing *tdcE* on a plasmid. We found that overexpression of *tdcE* in a Δ5 Δ*pflB* strain enabled fast growth with propionate (purple line in *Figure 8C*), indicating that TdcE can replace PFL both in catalyzing pyruvate cleavage to generate formate (as noted previously [*Sawers et al., 1998*]) and in catalyzing 2 KB production from propionyl-CoA and formate. Moreover, overexpression of *tdcE* in a WT strain fed with propionate-1-$^{13}$C resulted in a high proportion of labeled isoleucine: more than 70% without externally added formate and ~90% with externally added formate, both are considerably higher than without *tdcE* overexpression (*Figure 9*). Overexpression of *tdcE* in the Δ5 Δ*metA* strain resulted in 95% labeled isoleucine, the highest labeling we observed (*Figure 9*). Overall, these results confirm that TdcE can support in vivo 2 KB synthesis from propionyl-CoA in a comparable manner to that of PFL.

## Discussion

This study serves to demonstrate the selective advantage underground metabolism may offer in nature, providing metabolic redundancy and flexibility which allows for the best use of

environmental carbon sources (*Notebaart et al., 2014*). We demonstrated two metabolic pathways for the biosynthesis of isoleucine which are based on the activity of promiscuous enzymes. The first pathway, uncovered under aerobic conditions, is based on rerouting the metabolism of *O*-succinyl-L-homoserine from methionine biosynthesis toward the production of 2 KB. This represents a rather classic case of underground metabolism: a promiscuous reaction, the activity of which normally has a negligible physiological contribution, is enhanced by the rewiring of cellular metabolism under conditions which couple it to growth. Interestingly, a change in the steady-state concentration of a seemingly unrelated metabolite – cysteine – greatly enhances this promiscuous activity and enables a new biosynthetic route.

Under anaerobic conditions, we uncovered a pathway that relies on the highly expressed PFL (*Neidhardt et al., 1996*) to condense propionyl-CoA and formate to 2 KB. This route, unlike the aerobic cleavage of *O*-succinyl-L-homoserine, was highly active also in a WT strain, providing a substantial fraction of cellular isoleucine when propionate was present in the medium. As short-chain fatty acids, such as propionate, are abundant in the mammalian intestine (*McNeil et al., 1978*), the native habitat of *E. coli*, the KBFL pathway is likely to play a key role in isoleucine biosynthesis under physiological conditions. Hence, this pathway does not actually represent underground metabolism, but rather should be regarded an auxiliary biosynthesis route based on promiscuous enzyme activities. The use of the KBFL pathway is favorable over the canonical threonine cleavage route as it better uses the carbon sources available in the environment. As PFL supports the KBFL reaction under physiological conditions, we suggest renaming it 'pyruvate formate-lyase/2-ketobutyrate formate-lyase', in a similar manner to TdcE, which was named as '2-ketobutyrate formate-lyase/pyruvate formate-lyase'.

Several previous studies have suggested that, in various microbial lineages, 2 KB is derived from propionate via the activity of 2 KB:ferredoxin oxidoreductase (*Buchanan, 1969*; *Monticello et al., 1984*; *Eikmanns et al., 1983*; *Sauer et al., 1975*). The findings presented here indicate that, in at least some of these microorganisms, propionate might be converted to 2 KB via the KBFL route, where formate is produced internally via pyruvate cleavage, or even from the reduction of $CO_2$ via formate dehydrogenase (*Maia et al., 2017*). Even the microorganism in which the reductive carboxylation of propionyl-CoA was first demonstrated, that is, *Clostridium pasteurianum*, harbors a genomic PFL (*Kanehisa, 2000*) and thus may also assimilate propionate into isoleucine via the KBFL pathway. Moreover, in several microorganisms, propionyl-CoA is produced from succinyl-CoA as part of anaerobic fermentation (*Swick and Wood, 1960*; *Stams et al., 1984*; *Schink et al., 1987*; *Tholozan et al., 1994*); these organisms might be able to use the KBFL route without the need for an external supply of propionate. The prevalence of both the genes and the substrates involved in the KBFL pathway indicates that this route may be widespread.

Many cellular building blocks have alternative biosynthesis pathways. Nevertheless, isoleucine seems unusually rich in the sheer number of alternative routes for its bioproduction. There appear to be at least nine routes for 2 KB production: threonine cleavage (canonical pathway), the citramalate pathway (*Charon et al., 1974*), the glutamate mutase pathway (*Abramsky et al., 1962*), cystathionine cleavage (*Steegborn et al., 1999*), methionine cleavage (*Joshi and Jander, 2009*; *Joshi et al., 2010*), homolanthionine cleavage (*Krömer et al., 2006*), *O*-succinyl-L-homoserine cleavage (this study), propionyl-CoA reductive carboxylation (*Buchanan, 1969*; *Monticello et al., 1984*; *Eikmanns et al., 1983*), and propionyl-CoA condensation with formate (this study). The large repertoire of alternative pathways makes isoleucine biosynthesis the epitome of metabolic flexibility.

## Materials and methods

**Key resources table**

| Reagent type (species) or resource | Designation | Source or reference | Identifiers | Additional information |
|---|---|---|---|---|
| Strain, strain background (*Escherichia coli*) | *E. coli* strains | This study | | *Table 2* |

*Continued on next page*

**Table 2.** Strains used in this study.

| Strain name | Deletions/Genotype | Description | Source |
|---|---|---|---|
| BL21 (DE3) | *E. coli fhuA2 [lon] ompT gal (λ DE3) [dcm] ΔhsdS λ DE3 = λ sBamHIo ΔEcoRI-B int::(lacI::PlacUV5::T7 gene1) i21 Δnin5* | Strain for protein production | Lab collections |
| SIJ488 | *E. coli* K-12 MG1655Tn7:: pAra-exo-beta-gam; pRha-FLP; xylSpm-IsceI | MG1655 derivative with genome integrated recombinase and flippase genes | 57 |
| Δ*ilvA* Δ*tdcB* | SIJ488 Δ*ilvA* Δ*tdcB*::kan | Threonine deaminase deletion strain | This study |
| Δ5 | SIJ488 Δ*ilvA* Δ*tdcB* Δ*sdaA* Δ*sdaB* Δ*tdcG*::kan | Threonine deaminase and serine deaminase deletion strain | This study |
| Δ5 Δ*metA* | SIJ488 Δ*ilvA* Δ*tdcB* Δ*sdaA* Δ*sdaB* Δ*tdcG* Δ*metA*::kan | Threonine deaminase, serine deaminase and homoserine *O*-succinyltransferase deletion strain | This study |
| Δ5 Δ*metC* | SIJ488 Δ*ilvA* Δ*tdcB* Δ*sdaA* Δ*sdaB* Δ*tdcG* Δ*metC*::kan | Threonine deaminase, serine deaminase and cystathionine β-lyase deletion strain | This study |
| Δ5 *cysE*\* | SIJ488 Δ*ilvA* Δ*tdcB* Δ*sdaA* Δ*sdaB* Δ*tdcG* Δ*metC*::kan *cysE* | Threonine deaminase, serine deaminase and homoserine *O*-succinyltransferase deletion strain with *cysE* point mutation G97A | This study |
| Δ*ilvC* | SIJ488 Δ*ilvC*::cap | Ketol-acid reductoisomerase deletion strain | This study |
| Δ5 Δ*pflB* Δ*tdcE* | Δ*ilvA* Δ*tdcB* Δ*sdaA* Δ*sdaB* Δ*tdcG* Δ*pflB* Δ*tdcE*::cap | Threonine deaminase, serine deaminase, pyruvate formate-lyase and 2-ketobutyrate formate-lyase deletion strain | This study |
| Δ*pflB* | SIJ488 Δ*pflB*::kan | Pyruvate formate-lyase deletion strain | This study |
| Δ*tdcE* | SIJ488 Δ*tdcE*::cap | 2-Ketobutyrate formate-lyase deletion strain | This study |

*Continued*

| Reagent type (species) or resource | Designation | Source or reference | Identifiers | Additional information |
|---|---|---|---|---|
| Sequence-based reagent | | This study | PCR primers | **Supplementary file 2** |
| Recombinant DNA reagent | pZASS-tdcE (plasmid) | This study | | Overexpression vector of TdcE. |
| Recombinant DNA reagent | pCA24N-metB (plasmid) | This study | | Overexpression vector of MetB. |
| Recombinant DNA reagent | pCA24N-metC (plasmid) | This study | | Overexpression vector of MetC. |
| Recombinant DNA reagent | pCA24N-cysE (plasmid) | This study | | Overexpression vector of CysE. |
| Recombinant DNA reagent | pCA24N-cysE_A33T (plasmid) | This study | | Overexpression vector of CysE with a mutation A33T. |
| Recombinant DNA reagent | pORTMAGE | **Nyerges et al., 2016** | Addgene catalog no. 72680 | |
| Chemical compound, drug | $^{13}$C-1-glucose | Sigma Aldrich | | |

*Continued on next page*

*Continued*

| Reagent type (species) or resource | Designation | Source or reference | Identifiers | Additional information |
|---|---|---|---|---|
| Chemical compound, drug | $^{13}C$-3-glucose | Sigma Aldrich | | |
| Chemical compound, drug | Sodium propionate-1-$^{13}C$ | Sigma Aldrich | | |
| Commercial assay or kit | RNeasy Mini Kit | Qiagen | | |
| Commercial assay or kit | qScript cDNA Synthesis Kit | QuantaBio | | |
| Commercial assay or kit | Maxima SYBR Green/ROX qPCR Master Mix | Thermo Fisher Scientific | | |
| Software, algorithm | breseq | *Deatherage and Barrick, 2014* | RRID:SCR_010810 | |

## Strains and plasmids

All strains used in this study are listed in *Table 2*. The *E. coli* SIJ488 strain based upon K-12 MG1655 (*Jensen et al., 2016*) was used for the generation of deletion strains. SIJ488 is engineered to carry the gene deletion machinery in its genome (inducible recombinase and flippase). All gene deletions were carried out by successive rounds of λ-Red recombineering using kanamycin cassettes (FRT-PGK-gb2-neo-FRT [KAN], Gene Bridges, Heidelberg, Germany) or chloramphenicol cassettes (pKD3 [*Datsenko and Wanner, 2000*]) as described by *Baba et al., 2006*. Homologous extensions (50 bp) for the deletion cassettes were generated by PCR using oligonucleotides listed in *Supplementary file 2*.

For the overexpression of the 2-ketobutyrate lyase (*tdcE*), endogenous genes were amplified from *E. coli* genomic DNA using a two-step PCR (to remove cloning system relevant restriction sites [*Wenk et al., 2018*] – in this case a single site) using in the first round of PCR: primer pairs tdcE_-HIS_A + tdcE_B and tdcE_C + tdcE_CAP_D. The corresponding PCR products were combined in a second PCR using primer pairs tdcE_HIS_A and tdcE_CAP_D. The *tdcE* gene was subsequently cloned into cloning vector pNivC (*Wenk et al., 2018*) using restriction enzymes *Mph1103*I and *Xho*I, generating pNivC-tdcE. The *tdcE* gene was subsequently cloned from pNiv-tdcE into expression vector pZ-ASS using enzymes *Eco*RI and *Pst*I, resulting in pZASS-tdcE (*Wenk et al., 2018*).

Plasmids for the overexpression of MetB, MetC, and CysE (pCA24N-metB, pCA24N-metC, and pCA24N-cysE) were isolated from the ASKA collection (*Kitagawa et al., 2005*). The CysE Ala33Thr point mutation was generated by linear amplification of the pCA24N-cysE plasmid by PCR using following mismatch primers: CysE_A33T_fw and CysE_A33T_rv (*Supplementary file 2*). The template plasmid was digested by DpnI (Thermo Fisher Scientific, Darmstadt, Germany) at 37°C and *E. coli* DH5α was transformed with the reaction mixture to amplify the pCA24N-cysE_A33T plasmid. The mutation was confirmed by sequencing.

## Cultivation conditions

For strain maintenance, generation of deletion strains, and for growth during cloning LB medium (1% NaCl, 1% tryptone, 0.5% yeast extract) was used. Antibiotics for selection were used at the following concentrations: chloramphenicol, 30 µg/mL; kanamycin, 50 µg/mL; ampicillin, 100 µg/mL; streptomycin, 100 µg/mL. M9 minimal media was used for growth experiments (50 mM $Na_2HPO_4$, 20 mM $KH_2PO_4$, 1 mM NaCl, 20 mM $NH_4Cl$, 2 mM $MgSO_4$, 100 µM $CaCl_2$, 134 µM EDTA, 13 µM $FeCl_3 \cdot 6H_2O$, 6.2 µM $ZnCl_2$, 0.76 µM $CuCl_2 \cdot 2H_2O$, 0.42 µM $CoCl_2 \cdot 2H_2O$, 1.62 µM $H_3BO_3$, 0.081 µM, $MnCl_2 \cdot 4H_2O$) with carbon sources added to the concentrations specified in the text and figures.

For growth experiments, overnight cultures were incubated in 4 mL LB medium with no antibiotics for deletion strains, with strains having being previously plated on antibiotic media to ensure genotype. In the event a strain carried a plasmid, the relevant antibiotic was also added to the pre-culture medium. Before inoculation of the experiment, cultures were harvested and washed four times in M9 medium without carbon source by centrifugation (6000 rpm, 3 min) to remove residual carbon sources from the cells. Plate reader experiments were inoculated with a starting $OD_{600}$ of

0.01 using the washed culture. Plate reader experiments were carried out in 96-well microtiter plates (Nunclon Delta Surface, Thermo Fisher Scientific). Each well contained 150 µL of cell culture covered with 50 µL mineral oil (Sigma-Aldrich, Taufkirchen, Germany), to avoid evaporation. An Infinite M200 Pro plate reader (Tecan) was used for incubation (37°C), shaking, and $OD_{600}$ measurements. Three cycles of four shaking phases, each of 1 min were used (1. linear shaking at an amplitude of 3 mm, 2. orbital shaking at an amplitude of 3 mm, 3. linear shaking at an amplitude of 2 mm, and 4. orbital shaking at an amplitude of 2 mm). Optical density (OD 600 nm) was measured after each round of shaking (~12.5 min). Plate reader OD measurements were converted to cuvette values according to the formula $OD_{cuvette} = OD_{plate}/0.23$. Growth curves were processed in MATLAB and represent averages of (technical) triplicate measurements where the variability between triplicate measurements was less than 5% (unless explicitly stated, as in *Figure 2*). Within a given plate experiment, technical replicates averaged are replicates of a single genetic strain; biological replicates were generated for the Δ5 Δ*metC* strain and Δ5 *cysE** strains, these were engineered in parallel and were shown to behave in an identical manner.

Anaerobic growth experiments were performed in an Infinite M200 Pro plate reader (Tecan) inside a vinyl anaerobic chamber ($N_2$ with 10% $CO_2$, 2.5% $H_2$, model B, Coy Laboratory Products, Grass Lake, MI). M9 mediu for growth experiments were placed for at least 24 hr in the anaerobic chamber to allow for the exchange of dissolved oxygen from the media.

## Whole-genome sequencing

Genomic DNA was extracted using the NucleoSpin Microbial DNA kit (MACHERY-NAGEL, Düren, Germany) following manufacturer's instructions. Library construction and genome sequencing were performed by Novogene (Cambridge, United Kingdom) using the paired-end Illumina sequencing platform. Analysis of the sequencing data was performed using the *breseq* pipeline (*Deatherage and Barrick, 2014*) and the SIJ488 reference genome sequence, which is derived from the MG1655 (NC_000913, GenBank). *Supplementary file 1* shows the identified mutations, as compared to the parent strains.

## Multiplex automated genome engineering (MAGE)

Introducing genomic point mutations was achieved by using multiplex automated genome engineering (MAGE; *Wang et al., 2009*). A single colony of desired strain(s) transformed with pORTMAGE (*Nyerges et al., 2016*; Addgene catalog no. 72680) was incubated in LB medium supplemented with 100 mg $L^{-1}$ of ampicillin at 30°C in a shaking incubator. To start the MAGE cycle, overnight cultures were diluted by 100 times in the same medium and cultivated to an optical density of 0.4–0.5 at 600 nm. 1 mL of each culture was transferred to sterile microcentrifuge tubes, and then transferred to 42°C thermomixer (Thermomixer C, Eppendorf, Hamburg, Germany) to express λ-Red genes by heat shock for 15 min at 1000 rpm. After induction, cells were quickly chilled on ice for at least 15 min, and then made electrocompetent by washing three times with ice-cold ddH$_2$O. 40 uL of electrocompetent cell was mixed with 2 uL of 50 uM of oligomer stock solution and the final volume of the suspension was adjusted to 50 uL. The oligomer used to introduce the mutation was *CysE_A33T_MAGE* (*Supplementary file 2*). Electroporation was done on Gene Pulser XCell (Bio-Rad) set to 1.8 kV, 25 µF capacitance, and 200 Ω resistance for 1 mm gap cuvette. Immediately after electroporation, 1 mL of LB was added to cuvette and the electroporation mixtures in LB were transferred to sterile culture tubes and cultured with shaking at 30°C, 240 rpm for 1 hr to allow for recovery. After recovery, 2 mL of LB medium supplemented with ampicillin was added and then further incubated in the same condition. When the culture reached an $OD_{600}$ of 0.4–0.5, cells were either subjected to additional MAGE cycles or analyzed for genotype via PCR and sequencing.

## Isotopic-labeling of proteinogenic amino acids

Stationary isotope tracing experiments using $^{13}C$-1-labeled glucose, $^{13}C$-3-labeled glucose, and labeled sodium propionate (Sigma-Aldrich) were performed in order to understand the metabolic activities responsible for isoleucine biosynthesis. All experiments were performed in duplicate. Strains were grown in M9 with 10 mM labeled glucose as sole carbon source in the case of labeled glucose experiments, and 10 mM unlabeled glucose with 5 mM sodium propionate-1-$^{13}C$ in the case of the anaerobic experiments (extra carbon sources added where relevant at concentrations

indicated). Anaerobic experiments were performed also with an anaerobic pre-culture in M9 medium. The equivalent of 1 mL of culture at $OD_{600}$ one was harvested by centrifugation and washed twice in water. Cellular biomass was hydrolyzed by boiling in acid (95°C in 6 M HCl for 24 hr [*You et al., 2012*]). After drying the samples at 95°C, the samples were resuspended in ultra-pure water and the masses of amino acids analyzed with UPLC–ESI–MS as previously described (*Giavalisco et al., 2011*). Chromatography was performed with a Waters Acquity UPLC system (Waters), using an HSS T3 $C_{18}$ reversed phase column (100 × 2.1 $mm^2$, 1.8 µm; Waters). 0.1% formic acid in $H_2O$ (A) and 0.1% formic acid in acetonitrile (B) were the mobile phases. Flow rate was 0.4 mL/min and the gradient was: 0 to 1 min – 99% A; 1 to 5 min – linear gradient from 99% A to 82%; 5 to 6 min – linear gradient from 82% A to 1% A; 6 to 8 min – kept at 1% A; 8 to 8.5 min – linear gradient to 99% A; 8.5 to 11 min – re-equilibrate. Mass spectra were acquired using an Exactive mass spectrometer (Thermo Fisher Scientific) in positive ionization mode, with a scanning range of 50.0 to 300.0 m/z. Spectra were recorded during the first 5 min of the LC gradients. Retention times for amino acids under these conditions were determined by analyzing amino-acid standards (Sigma-Aldrich) under the same conditions. Data analysis was performed using Xcalibur (Thermo Fisher Scientific).

## Enzyme assays for MetB, MetC, and CysE

*E. coli* BL21 (DE3) was transformed with the overexpression plasmids pCA24N-metB, pCA24N-metC, pCA24N-cysE, or pCA24N-cysE_A33T, and grown on LB agar containing 34 µg/mL chloramphenicol. An expression culture in TB containing 34 µg/mL chloramphenicol was inoculated from the plate and grown at 37°C while shaking at 110 rpm until an $OD_{600}$ of roughly 0.9 was reached. The culture was shaken for another 4 hr at 37°C or overnight at 20°C and 110 rpm for gene expression. The culture was harvested by centrifugation at 4°C and 5000 *g*. Alternatively, the pellet was stored at −20°C. The pellet was resuspended in Buffer A (50 mM Tris-HCl pH 7.9, 500 mM NaCl) containing 10 µg/mL DNaseI and 5 mM $MgCl_2$. The cells were lysed by ultrasonication and the lysate was cleared by ultracentrifugation at 50,000 *g* and 4°C for 45 min followed by filtration through a 0.45 µm syringe filter. The lysate was loaded onto a 1 mL HisTrap FF column (GE Healthcare, Freiburg, Germany) equilibrated in Buffer A. Unspecifically bound protein was washed with Buffer A containing 50 mM imidazole. The protein was eluted with Buffer A containing 500 mM imidazole and subsequently desalted over a HiTrap 5 mL desalting column (GE Healthcare) using 20 mM Tris-HCl pH 7.9, 50 mM NaCl. The protein was concentrated using Amicon Ultra-4 centrifugal filters (Merck Millipore, Darmstadt, Germany). The protein was stored in 50% (v/v) glycerol at −20°C.

The MetB elimination reaction (red arrow, reaction one in *Figure 5*) was kinetically characterized using a spectrophotometric assay. The assays were performed at 30°C in 10 mm quartz cuvettes (Hellma Analytics, Germany) on a Cary-60 UV/Vis spectrometer (Agilent Technologies Inc, Santa Clara, CA). In the elimination reaction, MetB cleaves *O*-succinyl-L-homoserine to succinate and 2-ketobutyrate. The latter can be reduced by the coupling enzyme lactate dehydrogenase using NADH as reducing agent. The NADH consumption was followed at 340 nm ($\varepsilon$NADH = 6.22 $mM^{-1}$ $cm^{-1}$). The reaction was started by adding *O*-succinyl-L-homoserine to 22 nM MetB in 200 mM Tris-HCl pH 8.0 containing ~1 U of lactate dehydrogenase (Sigma-Aldrich Chemie GmbH, Germany) and 300 µM NADH. Kinetic parameters were determined from a Michaelis-Menten fit of 18 data points.

In the presence of *O*-succinyl-L-homoserine and cysteine or homocysteine, MetB can catalyze the replacement reaction yielding succinate and cystathionine or homolanthionine, respectively (MetB black arrow and reaction two in *Figure 5*). However, the presence of cysteine or homocysteine does not necessarily exclude the elimination reaction activity of the MetB enzyme (reaction one in *Figure 5*). To determine the reaction route, MetB was incubated with *O*-succinyl-L-homoserine and cysteine or homocysteine in separate experiments. To investigate the MetB reaction route at physiological cysteine or homocysteine concentrations, we ran an assay with 100 nM of MetB in 200 mM Tris-HCl pH 8.0 and 50 µM pyridoxal-5'-phosphate in the presence of 6 mM *O*-succinyl-L-homoserine and 300 µM of either cysteine or homocysteine. The assay mixture was incubated at 30°C and samples were quenched with 5% formic acid at specific time points. To investigate the MetB reaction route at high concentrations of cysteine and homocysteine, 90 nM of MetB was assayed in 200 mM Tris-HCl pH 8.0 in the presence of 6 mM *O*-succinyl-L-homoserine and either 3 mM L-cysteine or 6 mM L-homocysteine. The assay mixture was incubated at 30°C and samples were quenched with 10% formic acid at specific time points. In both experiments, the formation of 2 KB and succinate

was quantitatively determined by HPLC-MS/MS analysis. The levels of succinate shown in *Figure 5—figure supplement 1B–E* are the levels of succinate generated from the replacement reaction of MetB exclusively; the succinate generated by the elimination reaction has already been taken into account by subtracting the concentration of 2 KB in the sample (elimination reaction produces an equimolar amount of 2 KB and succinate).

To test whether MetC can produce 2 KB, we incubated 24 µM of MetC and 10 mM of homocysteine or 87 µM MetC and 33 mM of methionine in 50 mM Tris-HCl pH 8.5 at 30°C (reaction 5 or 6 in *Figure 5*). No 2 KB generation could be detected when following the NADH consumption of the coupling enzyme lactate dehydrogenase (~1 U of lactate dehydrogenase and 300 µM NADH). Some MetC variants from other organisms have also been reported to catalyze both the β-elimination (canonical *E. coli*) and the γ-elimination of cystathionine resulting in pyruvate and 2 KB, respectively. In this case, as both these products are accepted by the coupling enzyme lactate dehydrogenase (*Kim and Whitesides, 1988*), the reaction route could not be determined spectrophotometrically. Instead, 20 nM of MetC was incubated with 3 mM cystathionine in 50 mM Tris-HCl pH 8.5. The assay mixture was incubated at 30°C and samples were quenched with 10% formic acid at specific time points. Formation of 2 KB and pyruvate was quantitatively determined using HPLC-MS/MS analysis. Only pyruvate could be detected in these samples indicating no γ-elimination of cystathionine by the *E. coli* MetC enzyme.

The activity of CysE and its variant CysE Ala33Thr were assayed by photospectrometrically following the reaction of released CoA with 5,5′-dithiobis-(2-nitrobenzoic acid) (DTNB) at 412 nm using an extinction coefficient of 14.4 mM$^{-1}$ cm$^{-1}$. The assays were performed at 30°C in 10 mm quartz cuvettes (Hellma Analytics) on a Cary-60 UV/Vis spectrometer (Agilent Technologies). The reaction was started by adding the CysE variant to 0.2 mM DTNB in 100 mM potassium phosphate buffer pH 8. The dependence of the turnover number on acetyl-CoA concentration (synthesized as previously described [*Peter et al., 2016*]) was measured with 0.3 nM CysE or 1.3 nM CysE A33T at a saturating L-serine concentration of 20 mM. The dependence of the turnover number on L-serine concentration was measured with 1.0 nM CysE or 2.0 nM CysE A33T at a non-saturating acetyl-CoA concentration of 0.6 mM.

Succinate, pyruvate, and 2-ketobutyrate in the enzymatic assays were quantitatively determined using LC-MS/MS based on external calibration curves. The calibration curves were generated with at least seven concentrations within the linear range of the analyte in the matrix of the assay samples. The compounds were separated over a Kinetex EVO C18 chromatography column (50 × 2.1 mm$^2$; 3.5 µm, 100 Å, Phenomenex) equipped with a 20 × 2.1 mm$^2$ guard column of similar specificity on an Agilent Infinity II 1290 HPLC system. The column was heated to 55°C and 5 µL of sample or standard were injected. A constant flow rate of 0.2 mL/min was applied. Mobile phase A (0.1% formic acid in water [Honeywell, Morristown, New Jersey]) and mobile phase B (0.1% formic acid in acetonitrile [Honeywell]) were mixed according to the following mobile phase profile: 0–1.6 min constant at 0% B; 1.6–5 min from 0% to 95% B; 5–6 min constant at 95% B; 6–6.5 min from 95% to 0% B; 6.5–10 min constant at 0% B. An Agilent 6495 ion funnel mass spectrometer equipped with an electrospray ionization source in negative mode was used for analysis under the following conditions: ESI spray voltage 1500 V, sheath gas 200° C at 12 L/min, nebulizer pressure 30 psig, and drying gas 180° C at 11 L/min. Mass transition and retention time of the compounds were compared to standards for identification. The MassHunter software (Agilent, Santa Clara, CA, USA) was used for chromatogram integration.

## Anaerobic production of 2-ketobutyrate from PFL (i.e. PflB) and TdcE

Propionyl-CoA was synthesized and purified as described previously (*Peter et al., 2016*). The dried powder was kept at −20°C and dissolved in 50 mM acetate (pH 4.5) before use. The concentration was determined by two independent methods. First, a NanoDrop 2000 Spectrophotometer (Thermo Fisher Scientific) was used with the extinction coefficient of saturated CoA-esters at 260 nm ($\varepsilon_{260nm}$ = 16.4 mM$^{-1}$ cm$^{-1}$). Second, the depletion of the compound when incubated with the enzyme PduP (CoA-acylating propionaldehyde dehydrogenase; *Zarzycki et al., 2017*), following the oxidation of NADH at 340 nm ($\varepsilon_{340nm}$ = 6.22 mM$^{-1}$ cm$^{-1}$). Both methods gave the same results within a 5% error.

Overexpression plasmids for PFL-AE (PFL-activating enzyme), PFL, and TdcE were obtained from the ASKA collection. PFL-AE, PFL, and TdcE were produced in *E. coli* BL21(DE3). 1 L terrific broth

(TB) containing 34 µg/mL chloramphenicol was inoculated with freshly transformed cells and incubated at 37°C. After reaching an $OD_{600}$ of 0.8, the expression was induced by adding IPTG to a final concentration of 0.5 mM and the incubation temperature was lowered to 25°C. Cells were harvested after 16 hr by centrifugation (4500 $g$, 10 min) and resuspended in buffer A (50 mM HEPES-KOH pH 7.8, 500 mM KCl) containing 10 µg/mL DNaseI and 5 mM $MgCl_2$. If not used immediately, cell pellets were flash-frozen in liquid nitrogen and stored at −20°C. The cell lysate obtained by sonication was clarified by centrifugation 75,000 $g$ at 4°C for 45 min. The supernatant was filtered through a 0.4 µm syringe tip filter (Sarstedt). Ni-affinity purification was performed with an Äkta FPLC system from GE Healthcare. The filtered soluble lysate was loaded onto a 1 mL Ni-Sepharose Fast Flow column (HisTrap FF, GE Healthcare) that had been equilibrated with 10 mL buffer A. After washing with 20 mL 85% buffer A, 15% buffer B (50 mM HEPES-KOH pH 7.8, 500 mM KCl, 500 mM imidazole), the protein was eluted with 100% buffer B. Fractions containing purified protein were pooled and the buffer was exchanged to storage buffer (50 mM HEPES-KOH pH 7.8, 150 mM KCl) using a desalting column (HiTrap, GE Healthcare). Proteins were concentrated by ultrafiltration (Amicon Ultra). Protein concentration was determined on a NanoDrop 2000 Spectrophotometer (Thermo Fisher Scientific) using the extinction coefficient at 280 nm, as calculated by ProtParam (*Gasteiger, 2005*). Enzyme purity was confirmed using SDS-PAGE. The purified proteins were flash-frozen in liquid nitrogen and stored at −80°C.

PFL-AE was expressed and purified as PFL and TdcE. However, after the HisTrap purification step, the protein was transferred to an anaerobic glovebox. The reconstitution of the iron-sulfur cluster was performed following the protocol of *Broderick* et al. with a few adaptations (*Byer et al., 2018*). Using a PD-10 desalting column (GE Healthcare), the buffer was exchanged to an anaerobic reconstitution buffer (50 mM Tris-$SO_4$ pH 7.5, 150 mM KCl, 5 mM dithiothreitol [DTT]). The DTT was added fresh to the buffer right before use. After buffer exchange the protein was diluted with reconstitution buffer to a concentration of ~3 mg/mL (~100 µM) and transferred to a glass vial with a magnetic stirring bar. The vial was placed on ice on a magnetic stirrer. A 10 mM stock solution of $FeCl_3$ was added in 10 aliquots over the course of 1 hr to a final concentration of 600 µM. Then, a 50 mM stock of $Na_2S·(H_2O)_9$ was added in 10 aliquots over the course of 1 hr to a final concentration of 600 µM. This mixture was incubated on ice with stirring for 6 hr. Then the protein was desalted with a PD-10 column (equilibrated with reconstitution buffer), and concentrated on an Amicon to 30 mg/mL. UV-Vis spectra were recorded before and after the reconstitution with a Cary 4000 UV-Vis spectrometer (Agilent Technologies) using a FiberMate2 Fiber Optic Coupler (Harrick Scientific Products) with a path length of 1 cm (*Figure 8—figure supplement 2*). The reconstituted PFL-AE was transferred in aliquots to rubber stoppered HPLC vials, flash frozen in liquid nitrogen and stored at −80°C.

All anaerobic steps were performed in an anaerobic glovebox (Coy Laboratories) under an $N_2$ atmosphere containing 3% to 3.5% $H_2$. Residual $O_2$ was removed by palladium catalysts. $O_2$ concentration was monitored and maintained below 5 ppm at all times. All stocks were prepared under anaerobic conditions or prepared outside of the glovebox and equilibrated for at least 3 hr inside the glovebox. Dithionite and DTT stocks were freshly prepared under anaerobic conditions each day.

To activate PFL, 100 mM Tris-HCl pH 7.5, 50 mM KCl, 10 mM DTT, 50 mM sodium oxamate, 125 µM SAM, and 25 µM PFL-AE were incubated for 10 min at 30°C. Then, 25 µM PFL and 62.5 µM sodium dithionite were added. After 30 min at 30°C, the activation mix was placed on ice.

To activate TdcE, 100 mM Tris-HCl pH 7.5, 50 mM KCl, 10 mM DTT, 50 mM sodium oxamate, 50 µM SAM, and 10 µM PFL-AE were incubated for 10 min at 30°C. Then, 10 µM TdcE and 25 µM sodium dithionite were added. After 30 min at 30°C, the activation mix was placed on ice.

Glycyl radical formation was confirmed by exposing the activation mix to oxygen followed by SDS-PAGE analysis. The glycyl radical rapidly reacts with oxygen, cleaving off a ~3.5 kDa fragment from the C-terminus. PFL and TdcE were activated to a similar degree, as demonstrated in *Figure 8—figure supplement 2*.

Activation and stability of the activation mix was tested by the pyruvate cleavage assay (*Figure 8—figure supplement 2*). For this, 100 mM Tris-HCl pH 7.5, 50 mM KCl, 10 mM malate, 1 mM NAD, 20 mM pyruvate, 250 µM CoA, 4.5 U/mL citrate synthase from the porcine heart (Sigma-Aldrich), 27.5 U/mL malic dehydrogenase from the porcine heart (Sigma-Aldrich) were mixed and

the reaction initiated by adding the activation mix in 100-fold dilution. The reduction of NAD was followed at 340 nm.

2-Ketobutyrate formation kinetics were measured in a coupled enzyme assay: L-lactate dehydrogenase reduces 2-ketobutyrate to 2-hydroxybutyrate with the consumption of NADH. Absorption at 340 nm was recorded with a Cary 4000 UV-Vis spectrometer using a FiberMate2 Fiber Optic Coupler with a path length of 1 cm. Each substrate concentration was measured in triplicate. One activation mix was used up to 1.5 hr storage on ice, then a fresh activation mix was prepared for kinetic measurements. PFL and TdcE kinetic parameters take the half-of-the-sites activity (*Unkrig et al., 1989*) into account, that is, the dimer mass was used to calculate the protein concentration. Because the exact active site concentration (i.e. the glycyl radical content after activation) was not determined, $V_{max}$ is reported as specific activity. For Michaelis-Menten parameter calculation and fitting, GraphPad Prism 8 was used. For PFL with propionyl-CoA as substrate the data was fit to the substrate inhibition equation.

Propionyl-CoA kinetics: 100 mM Tris-HCl pH 7.5, 50 mM KCl, 500 mM sodium formate, 200 µM NADH, 25 U/mL L-lactate dehydrogenase from bovine heart (LDH; Sigma-Aldrich). The reaction was started by adding an activation mix in 100-fold dilution. Thus, the final concentration was 250 nM PFL and 100 nM TdcE, respectively. For PFL, the propionyl-CoA concentrations were 100 µM, 250 µM, 500 µM, 1 mM, 2.5 mM and for TdcE, 150 µM, 350 µM, 550 µM, 800 µM, 1.6 mM, 1.9 mM. In both cases, there was no activity in the absence of propionyl-CoA.

Formate kinetics: 100 mM Tris-HCl pH 7.5, 50 mM KCl, 200 µM NADH, 25 U/mL LDH. Propionyl-CoA was added to a final concentration of 800 µM (PFL) or 1.9 mM (TdcE). The reaction was started by adding an activation mix in 100-fold dilution. Thus, the final concentration was 250 nM PFL and 100 nM TdcE, respectively. For PFL, the formate concentrations were 12 mM, 40 mM, 100 mM, 250 mM, 500 mM, and for TdcE, 5 mM, 12 mM, 40 mM, 100 mM, 500 mM. In both cases, there was no activity in the absence of formate.

## Transcript level analysis by reverse transcriptase quantitative PCR

To determine mRNA levels, total RNA was extracted from three biological replicates from cells in exponential phase growing on M9 minimal medium with 10 mM glucose (supplemented with 2 mM isoleucine in case of the Δ5 strain). Total RNA was purified using the RNeasy Mini Kit (Qiagen, Hilden, Germany) as instructed by the manufacturer. In brief, ~$2.5 \times 10^8$ cells (0.5 mL of $OD_{600}$ 0.5) were mixed with 2 volumes of RNAprotect Bacteria Reagent (Qiagen) and pelleted, followed by enzymatic lysis, on-column removal of genomic DNA with RNase-free DNase (Qiagen) and spin-column-based purification of RNA. Concentration and integrity of the isolated RNA were determined by NanoDrop and gel electrophoresis. cDNA was synthesized via reverse transcription of 500 ng RNA with the qScript cDNA Synthesis Kit (QuantaBio, Beverly, MA). Quantitative real-time PCR was performed two times in technical triplicates using the Maxima SYBR Green/ROX qPCR Master Mix (Thermo Fisher Scientific). An input corresponding to 25 pg total RNA/cDNA was used per reaction. Non-specific amplification products were excluded by melting curve analysis. The gene encoding 16S rRNA (*rrsA*) was chosen as a well-established reference transcript for expression normalization (*Zhou et al., 2011*). Two primer pairs for amplification of *metA* and *metB*, respectively, were used for qPCR (*Supplementary file 2*). Equal amplification efficiencies between the primers for the genes of interest and the reference gene were assumed. Differences in expression levels were calculated according to the $2^{-\Delta\Delta Ct}$ method (*Livak and Schmittgen, 2001*). Reported data represents the average of $2^{-\Delta\Delta Ct}$ values that were calculated for each sample individually relative to the average of all biological WT replicate $\Delta Ct^{(Ct(GOI)-Ct(rrsA))}$ values. Negative control assays with the direct input of RNA (without previous reverse transcription) confirmed that the observed fold change of *metA* and *metB* in the Δ5 *cysE** sample was not due to an increase in genomic DNA contamination, that is, it accounted for less than ~10% of the signal (ΔCt between +RT/–RT samples >3).

## Cellular concentrations of methionine and succinyl-homoserine

To assess the relative intracellular concentration changes of succinyl-homoserine and methionine among different strains, we cultivated the strains in four independent replicates of 20 mL media in 100 mL Erlenmeyer flasks (n = 4) to mid-exponential phase ($OD_{600}$ approx. 0.4–0.8). After measuring $OD_{600}$, 12 mL culture were immediately harvested and quenched with 36 mL precooled 60%

methanol (v/v). Cells were spun down by centrifugation (Beckman Allegra 25R refrigerated centrifuge): 5000 *g*, −10℃, 10 min, and stored at −80℃ before extraction. On ice, the cells were added to precooled extraction fluid (10 mM Tris-HCl, 1 mM EDTA, 50% MeOH (v/v), pH 7.0) and chloroform 0.5 mL of each, briefly vortexed and shaken at 4℃ for 2 hr for extraction. After centrifugation (5000 *g*, −10℃, 10 min), the upper phase of the extracts were filtered (Fisher syringe filter, PTFE, 0.2 µM, 13 mm diameter), and stored at −80℃ until further processing.

Quantitative determination was performed using an HRES-LC-MS. The chromatographic separation was performed on a Vanquish HPLC system using a ZicHILIC SeQuant column ($150 \times 2.1$ mm$^2$, 3.5 µm particle size, 100 Å pore size) connected to a ZicHILIC guard column ($20 \times 2.1$ mm$^2$, 5 µm particle size) (Merck KgAA) with a constant flow rate of 0.3 mL/min with mobile phase A being 0.1% formic acid in 99:1 water:acetonitrile (Honeywell) and phase B being 0.1% formic acid 99:1 water: acetonitrile (Honeywell) at 45° C.

The injection volume was 2 µL. The mobile phase profile consisted of the following steps and linear gradients: 0–1 min constant at 90% B; 1–8 min from 90% to 20% B; 8–9 min constant at 20% B; 9–9.1 min from 20% to 90% B; 9.1–11 min constant at 90% B. An ID-X Orbitrap mass spectrometer equipped with a HESI electrospray ion source (Thermo Fisher Scientific) was used at the following conditions: ESI spray voltage 3500 V, sheath gas at 50 AU, auxiliary gas at 10 AU and sweep gas at 1 AU, with the vaporizor temperature being 350℃ and the ion transfer tube temperature being 325° C. Compounds were identified based on their accurate mass and retention time compared to standards. Chromatograms were integrated using Xcalibur software (Thermo Fisher Scientific). Absolute concentrations were calculated based on an external calibration curve.

## Relative cellular concentrations of cysteine and homocysteine

The concentration of cysteine could not be determined using the above method due to its instability during the measurement process. Therefore, we used another method to assess the relative intercellular concentration of these amino acids.

To assess the relative intracellular concentration changes of cysteine, homocysteine, and 2 KB among different strains, we cultivated the strains in four independent replicates of 4 mL media in glass test tubes (n = 4) to mid-exponential phase (OD$_{600}$ approx. 0.4–0.8). About $10^9$ cells (equivalent of 1 mL of OD$_{600}$ = 1 culture) were collected on 0.45 µm Durapore membrane filters (Merck Millipore, Ireland). After washing the filters with 1 mL of fresh medium in under 10 s, the filters with cells were placed in 5 mL of −20℃ 40:40:20 (v/v/v) acetonitrile/methanol/water overnight. 4 mL of debris-free extract per sample was dried by vacuum centrifugation at 35℃ overnight. The dried metabolites were stored at −20℃ until further processing.

Metabolites were methoxyaminated and trimethylsilylated before gas chromatography-atmospheric pressure chemical ionization-quadrupole time of flight mass spectrometry (GC/APCI-qTOF-MS) as described earlier (*Kopka et al., 2017*). GC/APCI-qTOF-MS analyses were performed with an Agilent 7890B gas chromatograph (Agilent Technologies Deutschland GmbH, Germany) hyphenated to a Bruker Impact II mass spectrometer (Bruker Daltonik GmbH, Bremen, Germany). The machine parameters and analysis procedures were exactly as detailed in *Kopka et al., 2017*. Data mining was performed with Profile Analysis Version 2.2 software (Bruker Daltonik GmbH).

## Acknowledgements

This study was funded by the Max Planck Society and by the German Federal Ministry of Education amd Research, Grant no. 031B0194 (project FormatPlant) .

## Additional information

### Funding

| Funder | Grant reference number | Author |
|---|---|---|
| Max Planck Society | | Charles AR Cotton<br>Iria Bernhardsgrütter<br>Hai He<br>Simon Burgener<br>Luca Schulz<br>Nicole Paczia<br>Beau Dronsella<br>Alexander Erban<br>Stepan Toman<br>Marian Dempfle<br>Alberto De Maria<br>Joachim Kopka<br>Steffen N Lindner<br>Tobias J Erb<br>Arren Bar-Even |
| Federal Ministry of Education and Research | 031B0194 (FormatPlant) | Charles AR Cotton<br>Simon Burgener<br>Luca Schulz<br>Stepan Toman<br>Steffen N Lindner<br>Tobias J Erb<br>Arren Bar-Even |

The funders had no role in study design, data collection and interpretation, or the decision to submit the work for publication.

### Author contributions

Charles AR Cotton, Conceptualization, Data curation, Formal analysis, Investigation, Visualization, Methodology, Writing - original draft, Writing - review and editing; Iria Bernhardsgrütter, Formal analysis, Investigation, Visualization, Methodology; Hai He, Nicole Paczia, Alexander Erban, Stepan Toman, Marian Dempfle, Joachim Kopka, Steffen N Lindner, Formal analysis, Investigation, Methodology; Simon Burgener, Luca Schulz, Alberto De Maria, Investigation, Methodology; Beau Dronsella, Formal analysis, Validation, Investigation, Methodology; Tobias J Erb, Supervision; Arren Bar-Even, Conceptualization, Supervision, Funding acquisition, Investigation, Visualization, Writing - original draft, Project administration, Writing - review and editing

### Author ORCIDs

Iria Bernhardsgrütter (iD) http://orcid.org/0000-0002-5019-8188
Hai He (iD) http://orcid.org/0000-0003-1223-2813
Simon Burgener (iD) https://orcid.org/0000-0003-3703-168X
Joachim Kopka (iD) http://orcid.org/0000-0001-9675-4883
Steffen N Lindner (iD) http://orcid.org/0000-0003-3226-3043
Arren Bar-Even (iD) https://orcid.org/0000-0002-1039-4328

### Decision letter and Author response

Decision letter https://doi.org/10.7554/eLife.54207.sa1
Author response https://doi.org/10.7554/eLife.54207.sa2

## Additional files

### Supplementary files

• Supplementary file 1. Summary of identified mutations in 16 strains lacking threonine deaminases ($\Delta ilvA$ $\Delta tdcB$) or threonine and serine deaminases ($\Delta ilvA$ $\Delta tdcB$ $\Delta sdaA$ $\Delta sdaB$ $\Delta tdcG$) that were evolved to grow on 10 mM glucose after 70 hr without isoleucine or 2 KB.

• Supplementary file 2. List of all primers used in this study.

- Transparent reporting form

### Data availability
All data generated or analysed during this study are included in the manuscript and supporting files. Source data files have been provided for Figures 2 and 7 as well as for the metabolomic analysis.

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
