## [Decision Letter]

**Acceptance summary:**

This work has expanded the understanding of isoleucine biosynthesis. It highlights flexibility and interconnectivity of main and alternative metabolic pathways. The knowledge should be useful for further development in the fields of systems biology and metabolic engineering.

**Decision letter after peer review:**

Thank you for submitting your article "Underground isoleucine biosynthesis pathways in *E. coli*" for consideration by *eLife*. Your article has been reviewed by three peer reviewers, one of whom is a member of our Board of Reviewing Editors, and the evaluation has been overseen by Michael Marletta as the Senior Editor. The following individual involved in review of your submission has agreed to reveal their identity: Yew Wen Shan (Reviewer #2).

The reviewers have discussed the reviews with one another and the Reviewing Editor has drafted this decision to help you prepare a revised submission.

Summary:

The manuscript by Cotton et al. presents mechanisms underlying microbial adaptation to evolve alternative rescue pathways when the main Isoleucine biosynthesis pathway in *E. coli* is disrupted. Cells in which threonine deaminases have been deleted are still able to grow, implying that another source of 2-ketobutyrate for synthesis of isoleucine must be available. The authors demonstrated that within the proto-typical microbe, *Escherichia coli*, isoleucine auxotrophy can be complemented/compensated by multiple pathways. As the disruption was carried out at the step of 2-ketobutyrate production, the cell uses other enzymes, MetB mainly under aerobic where the provision of the critical metabolite is done by succinyl-homoserine cleavage, and pyruvate format lyase (PflB/TdcE) to provide 2KB under anaerobic conditions by the condensation of propionyl-CoA.

Overall, the work is well written and the results are quite interesting. The authors make a good case that the enzymes responsible for rescuing growth differ when the cells are grown under aerobic vs anaerobic conditions. The suggestion that a significant amount of 2-ketobutyrate may be produced from propionate in the anaerobic conditions in the mammalian gut is particularly intriguing. The work contributes an elaboration of "underground metabolism" where latent metabolic pathways catalyzed by promiscuous enzymes can be activated given the right metabolic/physiological conditions.

However, there are many issues need to be addressed clearly (especially the issues mentioned in the first two comments). Otherwise, the manuscript would not be accepted for publication in *eLife*.

Essential revisions:

1) The authors report that replicate cultures of *∆ilvA∆tdcBE. coli*, which lack both threonine deaminases, begin to grow at different times, yet genome sequencing reveals no mutations. The authors claim that the is due to "stochastic adaptation of cellular metabolism". This doesn't make sense; typically this kind of result is seen when mutations are required, and they occur at different times in different tubes. Transcriptional changes in response to growth conditions typically happen very quickly, and within the same time frame for replicate cultures. Further, the methionine synthesis genes should be turned on already in minimal medium, so it is not clear what other changes would be contributing. This is a major problem with either the data or the interpretation that needs to be rectified.

2) According to Ecocyc, MetC is an essential gene and strains lacking MetC cannot grow on glucose as a sole carbon source. Thus, the observation that the ∆5 *∆metC* strain can grow without addition of methionine is puzzling. Did the authors confirm that metC is really absent? Either there is an interesting and non-obvious mechanistic explanation, or the work was not done properly.

3) It is odd that cells grown under aerobic conditions can recruit MetB to synthesize 2-ketobutyrate, while cells grown under anaerobic conditions apparently do not. MetB should be expressed under both conditions. A good explanation for this counter-intuitive finding is necessary.

4) The text in several places says that enzyme x can catalyze reaction y. For example, subsection “2-ketobutyrate biosynthesis from succinyl-homoserine” says MetB catalyzes both condensation of o-succinylhomoserine and cysteine and cleavage of o-succinylhomoserine to succinate and 2-ketobutyrate. In every case where such statements are made, it is important to provide values for *k_cat_* and *K_m_*, either from the literature or from experiments carried out in the course of this work.

5) Actual values of K_i_ for the inhibition of o-succinylhomoserine cleavage by MetB in the presence of cysteine and homocysteine should be determined. These values should be compared to the actual concentrations of cysteine and homocysteine in cells. The single concentrations used in the experiments shown in Supplementary Figure 1B are far above the concentrations that I would expect to be present in cells.

6) Subsection “A latent aerobic isoleucine biosynthesis pathway” paragraph four: Is it ok to ignore flux through the pentose phosphate pathway when considering labeling patterns after growth on 13C-labelled glucose? Considerable flux goes through it.

7) The postulated propionate formate lyase activity of pyruvate formate lyase should be quantified in vitro.

8) Results section and Figure 3, in order to establish that the lack of 2-ketobutyrate is a major cause for the phenotype observed in Figure 3. A simple experiment such as adding 2-ketobutyrate, not isoleucine, into the growth medium to identify if adding 2-ketobutyrate gives the same effect as adding isoleucine would be helpful to confirm the role of 2-ketobutyrate in the mutants.

9) Subsection “2-ketobutyrate biosynthesis from succinyl-homoserine”: Why did the authors not construct the mutant in which MetB was deleted to confirm the conclusion made about MetB?

10) Subsection “Enzyme assays”: MetB assay needs clearer description about product measurement. The ability of MetB to generate 2-ketobutyrate should be measured by HPLC/MS (which the authors did with the inhibition experiments). For MetB kinetic assays, the authors described the measurement by monitoring NADH consumption. It is not clear how the assay is linked to 2-ketobutyrate formation.

11) Subsection “Enzyme assays” describes the assay used for MetB activity. The MetB elimination reaction they are considering produces 2-ketobutyrate, which is apparently being detected with lactate dehydrogenase, which normally uses pyruvate as a substrate. Does lactate dehydrogenase turn over 2-ketobutyrate? Did the authors ensure that the amount of lactate dehydrogenase was sufficient so that they were actually assaying MetB activity? Paragraph five says that "succinate, pyruvate and 2-ketobutyrate in the enzymatic assays were quantitatively determined by LC/MS/MS.” This statement is not consistent with the described use of lactate dehydrogenase.

12) In the presence of substantial genetic, growth and complementation assays, the work should benefit from the in vitro enzymatic demonstration of activity of TdcE and PflB.

[Editors' note: further revisions were suggested prior to acceptance, as described below.]

Thank you for resubmitting your work entitled "Underground isoleucine biosynthesis pathways in *E. coli*" for further consideration by *eLife*. Your revised article has been evaluated by Michael Marletta (Senior Editor) and a Reviewing Editor.

The manuscript has been improved but there are some remaining issues that need to be addressed before acceptance, as outlined below:

All reviewers think that the manuscript is very much improved and presents convincing evidence for the existence of alternative pathways for synthesis of 2KB. However, one of the reviewers note that a solid explanation for how the mutations increase flux through alternative pathways for 2KB synthesis should be provided because this is an important missing piece of the puzzle.

Essential revisions:

1) The mechanism by which the mutation in CysE improves synthesis of 2KB is puzzling. It seems logical that decreasing the concentration of cysteine would increase the rate of cleavage of O-succinyl-L-homoserine to 2KB and succinate. However, the mutation appears to decrease the cysteine concentration by only 44%. (There is enough variability in the data that this is not a totally convincing finding; the p value for the difference between the ∆5 and the ∆5 *cysE** strains is 0.16.) In other data, the authors show that, in the presence of cysteine, O-succinyl-L-homoserine is converted entirely to cystathionine and succinate (the normal intermediates in the methionine synthesis pathway). Therefore, it would not be expected that O-succinyl-L-homoserine is cleaved to 2KB and succinate in vivo in the presence of a substantial amount of cysteine. Can the authors come up with a good explanation? For example, is it possible that both the CysE mutation and the *metC* deletion lower methionine levels, which then would lead to increased transcription of *metB*? Decreasing MetC activity might also lead to an increase in the concentration of O-succinyl-L-homoserine, which could help push material toward 2KB synthesis. It would be nice if this could be verified by metabolomics.

2) Gels showing the purity of isolated enzymes should be included in the supplementary material.

3) Subsection “2-ketobutyrate biosynthesis from succinyl-homoserine” final paragraph and legend to Figure 5—figure supplement 1 – it is not correct to say that cysteine inhibits MetB. Cysteine is a substrate for MetB. It doesn't really inhibit cleavage of O-succinyl-L-homoserine to 2KB and succinate. It's just that the intermediate formed by reaction of O-succinyl-L-homoserine with the PLP cofactor at the active site of the enzyme is directed toward a different fate in the absence of cysteine.

4) In subsection “Disruption of MetC or a mutation in serine acetyltransferase enable steady 2-ketobutyrate production from succinyl-homoserine” – the text reports an apparent *k_cat_* for CysE and A33T CysE. According to the Materials and methods section, these enzymes were assayed with only a single concentration of substrates (20 mM serine and 0.2 mM acetyl CoA). In the absence of information about the *K_m_* for each substrate for the wild-type and mutant enzymes, it is not obvious that the values measured are truly *k_cat_*, particularly if the Ala33Thr change has a significant effect on a *K_m_*. (This is mostly a concern for acetyl CoA, since the concentration of serine was quite high.) Also, for the purpose of interpreting the effect of the mutation, it is necessary to determine whether the enzyme is saturated in vivo (i.e. whether it is *k_cat_* or *k_cat_* / *K_m_* that is the physiologically relevant parameter),

5) Subsection “Anaerobic 2KB biosynthesis from a reversible 2KB formate-lyase activity”: – activity should be given in terms of *k_cat_* rather than specific activity.

6) The KBFL pathway appears to be a major pathway for production of 2KB under anaerobic conditions. Therefore, it isn't really an underground pathway. Maybe it should be called an auxiliary pathway?

7) Proper terminology for the chemical intermediates should be used (e.g. succinyl homoserine should be O-succinyl-L-homoserine).

8) Subsection “Enzyme assays for MetB, MetC, and CysE” paragraph four: – a reference should be provided for the statement that lactate dehydrogenase can reduce 2KB.

9) Table 2 – *∆ilvA ∆tdcB* should be threonine deaminase deletion strain.

---

## [Author Response]

Essential revisions:1) The authors report that replicate cultures of ∆ilvA ∆tdcB *E. coli*, which lack both threonine deaminases, begin to grow at different times, yet genome sequencing reveals no mutations. The authors claim that the is due to "stochastic adaptation of cellular metabolism". This doesn't make sense; typically this kind of result is seen when mutations are required, and they occur at different times in different tubes. Transcriptional changes in response to growth conditions typically happen very quickly, and within the same time frame for replicate cultures. Further, the methionine synthesis genes should be turned on already in minimal medium, so it is not clear what other changes would be contributing. This is a major problem with either the data or the interpretation that needs to be rectified.

The reviewers are indeed correct and we thank them for noting this problem. Following this comment, we decided to sequence the genome of more strains that started growing after >70 hours. We used another software to identify mutations. We found, in multiple strains, the deletion of a large genomic sequence (>25,000 bp) that we were not able to identify using the software we previously used. Specifically, we found that in most of the sequenced strains (11 strains out of the 16 sequenced) *metC* was either deleted or mutated. In only 2 strains we were not able to identify any mutation, but this might be due to technical difficulties.

We provide the sequencing results in a new Supplementary Table. We removed any indication to "stochastic adaptation of cellular metabolism" from the text. We further added the following text to the Results section:

“Next, we aimed to understand the genetic basis underlying 2KB biosynthesis in the Δ*ilvA* Δ*tdcB* and Δ5 strains. […] Another mutated strain harbored a single mutation in the gene coding for serine acetyltransferase (CysE): Ala33Thr (Supplementary file 1)…”

2) According to Ecocyc, MetC is an essential gene and strains lacking MetC cannot grow on glucose as a sole carbon source. Thus, the observation that the ∆5 ∆metC strain can grow without addition of methionine is puzzling. Did the authors confirm that metC is really absent? Either there is an interesting and non-obvious mechanistic explanation, or the work was not done properly.

We acknowledge that this result was originally quite surprising. To confirm that the *metC* gene was absent we performed:

i) PCR with “external” primers for the region of the deleted gene, confirming that it is not in its original position.

ii) PCR with “internal” primers, confirming that the gene cannot be found anywhere in the genome (that is, that the gene did not “jump” to any other genomic location).

Importantly, despite the EcoCyc annotation, there are multiple indications in the literature that *metC* can be deleted with no auxotrophic phenotype:

i) In the Keio collection publication, it is stated that Δ*metC* grows on MOPS + 0.4% glucose (see supplementary table 3 in Baba, Tomoya, et al. "Construction of *Escherichia coli* K<inline-graphic mime-subtype="png" mimetype="image" xlink:href="media/image1.png" />12 in<inline-graphic mime-subtype="png" mimetype="image" xlink:href="media/image1.png" />frame, single <inline-graphic mime-subtype="png" mimetype="image" xlink:href="media/image2.png" />gene knockout mutants: the Keio collection." Molecular systems biology 2.1 (2006).

ii) One study clearly showed that MalY can replace MetC (Zdych et al., 1995).

iii) Another study showed that Alr and FimE can replace MetC (Patrick et al., 2007).

Taken together, it is clear that several PLP-dependent enzymes can replace MetC, explaining the phenotype we found, which is in line with previous studies.

Following the reviewers’ comment, we now cite these studies in the revised text and further explain that the promiscuous activity of PLP-dependent enzymes can explain growth without MetC:

“The Δ5 *ΔmetC* strain could grow even without the addition of methionine (which is in line with the *ΔmetC* strain in the Keio collection), presumably due the existence of multiple PLP-dependent enzymes that can catalyze MetC reaction, e.g., MalY, Alr, and FimE.”

Finally, we note that in multiple of the strains evolved to grow without isoleucine, a large genomic sequence, containing the *metC* gene, was deleted, further confirming that *E. coli* can grow without MetC activity.

3) It is odd that cells grown under aerobic conditions can recruit MetB to synthesize 2-ketobutyrate, while cells grown under anaerobic conditions apparently do not. MetB should be expressed under both conditions. A good explanation for this counter-intuitive finding is necessary.

We thank the reviewers for raising this important point. As indicated by our study, overexpression of MetB is probably not the correct approach to test MetB-dependent production of 2KB, due to the inhibition of the reaction by cysteine. Instead, we tested the evolved Δ*ilvA* Δ*tdcBcysE** strain and the constructed Δ5 *cysE** strain anaerobically to see whether the MetB-dependent route is active also under these conditions. As shown in the newly added Figure 7—figure supplement 1, we found that *cysE** indeed enables growth also under anaerobic conditions but with a very different growth profile to the propionate dependent growth. It is worth noting that while this mutation enabled growth, it did not arise spontaneously under anaerobic conditions.

One could also infer that the MetB-dependent pathway is active anaerobically by comparing the labeling of isoleucine upon feeding with ^13^C-propionate of the ∆5 strain and the ∆5 ∆*metA* strain in Figure 8. The increased labeling observed upon deletion of *metA* indicates that the MetB-dependent route is active anaerobically although to a much lower extent than aerobically.

The revised text reflects these points:

“Next, we aimed to explore underground isoleucine biosynthesis routes under anaerobic conditions. The evolved Δ*ilvA* Δ*tdcBcysE** strain and the constructed Δ5 *cysE** strain could grow anaerobically without isoleucine (Figure 8—figure supplement 1). However, we did not observe the reemergence of the *metC* deletion, the *cysE** mutation, or any other mutation: the Δ5 strain failed to grow without the addition of isoleucine, even after 120 hours (black lines in Figure 8A).”

“Deletion of the metA increased this labeling to 90% (Figure 9), indicating that the MetB-dependent route is active anaerobically although to a much lower extent than aerobically.”

4) The text in several places says that enzyme x can catalyze reaction y. For example, subsection “2-ketobutyrate biosynthesis from succinyl-homoserine” says MetB catalyzes both condensation of o-succinylhomoserine and cysteine and cleavage of o-succinylhomoserine to succinate and 2-ketobutyrate. In every case where such statements are made, it is important to provide values for k_cat_ and K_m_, either from the literature or from experiments carried out in the course of this work.

Following the reviewer’s, we added the required information:

“First, rather than catalyzing the condensation of cysteine with succinyl-homoserine – the first intermediate in the methionine biosynthesis pathway – cystathionine γ-synthase (MetB) can cleave the latter intermediate to succinate and 2KB (reaction 1 in Figure 5; *k_cat_* = 7.7 s^-1^ and *K_M_*(succinyl-homoserine) = 0.33 mM for *E. coli* enzyme). […] Finally, some MetC variants can act as methionine γ-lyase, releasing 2KB by directly cleaving methionine (reaction 6 in Figure 5; e.g., *k_cat_* ≈ 0.01 s^-1^ for *Lactococcus lactis* subsp. *cremoris* B78 enzyme).”

“Supporting previous studies, we found MetB to catalyze the cleavage of succinyl-homoserine to 2KB (reaction 1 in Figure 5) with a *k_cat_* of 9.3 ± 0.4 s-1, *K_m_* (succinyl-homoserine) = 0.60 ± 0.08 mM and k_cat_/ *K_m_* (succinyl-homoserine) = 16 ± 2 mM^-1^ s^-1^ (Figure 5—figure supplement 2A)”

“We indeed found that PFL catalyzes the condensation of propionyl-CoA and formate to give 2KB with a specific activity of 0.6-1.1 μmol/min/mg with *K_m_* (propionyl-CoA) = 0.83 ± 0.6 mM and *K_m_* (formate) = 69 ± 11 mM (Table 1, Materials and methods, and Figure 5—figure supplement 1, and Figure 8—figure supplement 2 and 3).”

“We found that purified TdcE is catalytically superior to PFL generating 2KB with a specific activity of 1.8-2.1 μmol/min/mg with *K_m_* (propionyl-CoA) = 0.52 ± 0.08 mM and *K_m_* (formate) = 20 ± 3 mM (Table 1, Materials and methods, and Figure 5—figure supplement 1, and Figure 8—figure supplement 2 and 3).”

5) Actual values of K_i_ for the inhibition of o-succinylhomoserine cleavage by MetB in the presence of cysteine and homocysteine should be determined. These values should be compared to the actual concentrations of cysteine and homocysteine in cells. The single concentrations used in the experiments shown in Supplementary Figure 1B are far above the concentrations that I would expect to be present in cells.

We thank the reviewers for this very important comment. Reviewing the literature, we find consistent evidence that *E. coli*‘s intracellular concentration of both cysteine and homocysteine is ~0.3 mM (Wheldrake, 1967; Bennett et al., 2009; Guo et al., 2013).

Therefore, we performed additional experiments where we incubated the enzyme with Osuccinylhomoserine and physiological concentration of cysteine or homocysteine at 0.3 mM. We followed the two reactions by quantitatively determining the concentrations of succinate and 2KB over time by HPLC-MS/MS. When MetB is incubated with O-succinylhomoserine and 0.3 mM cysteine, the elimination reaction (reaction 1 in Figure 5) is completely inhibited as long as cysteine is present and only begins once all the cysteine is used up. Homocysteine (0.3 mM) seems to have a weaker inhibitory effect on the cleavage reaction. All relevant graphs are presented in Supplementary Figures 1B-E and hopefully give a clearer picture of how cysteine and homocysteine work in repressing the elimination reaction of this enzyme.

Unfortunately, it is impossible to measure K_i_ values for cysteine and homocysteine for the inhibition of O-succinylhomoserine cleavage to succinate and 2KB by MetB (reaction 1 in Figure 5). Cysteine and homocysteine can both be used as substrates themselves in the MetB replacement reactions (the standard reaction marked by the black arrow in Figure 5 and the 2KB producing reaction labelled 2 in Figure 5 respectively) and for cysteine, it has been reported that the replacement reaction is performed with a 65fold increased turnover rate compared to the elimination reaction (Aitken, S. M., Kim, D. H. & Kirsch, J. F. (2003) *Escherichia coli* cystathionine γ-synthase does not obey ping-pong kinetics. Novel continuous assays for the elimination and substitution reactions, *Biochemistry.* 42, 11297-306). This makes it impossible to investigate the inhibitory effect of cysteine and homocysteine, particularly at low concentrations as they are converted quickly.

We have amended the text to reflect the new results:

“Supporting previous studies, we found MetB to catalyze the cleavage of succinyl-homoserine to 2KB (reaction 1 in Figure 5) with *k_cat_* = 9.3 ± 0.4 s^-1^, *K_m_* (succinyl-homoserine) = 0.60 ± 0.08 mM, and thus *k_cat_*/ *K_m_* (succinyl-homoserine) = 16 ± 2 mM^-1^ s^-1^ (Figure 5—figure supplement 2A). Yet, it was previously reported that cysteine inhibits succinyl-homoserine cleavage. We therefore characterized the cleavage reaction in the presence of either cysteine or homocysteine (as an alternative substrate), at physiological concentrations of ~ 0.3 mM and at artificially high concentrations of 3-6 mM. We found that cysteine indeed inhibits the cleavage of succinyl-homoserine at physiological concentrations, while homocysteine inhibited the reaction only at high concentrations (Figure 5—figure supplement 2B-E).”

6) Subsection “A latent aerobic isoleucine biosynthesis pathway” paragraph four: is it ok to ignore flux through the pentose phosphate pathway when considering labeling patterns after growth on 13C-labelled glucose? Considerable flux goes through it.

Flux via the oxidative pentose phosphate pathway is indeed substantial but its effect on the labelling will depend on whether the resulting pentose phosphates are further converted to C3/C6 phosphosugars. That is, production of only pentose phosphates from the oxidative pentose phosphate will not change the labelling in pyruvate, while only if a substantial amount of pentose phosphates are converted to F6P and GAP, the labelling of pyruvate will be affected. The deviation of the measured labelling from the expected one also reflects this flux. Regardless, the take home message from this section is that the labelling patterns of the WT strain, the *ΔilvA ΔtdcB* strain, and the Δ5 strain are identical, indicating that the citramalate pathway does not operate.

We amended the text to indicate that the deviation from the expected labelling can be attributed also to the activity of the pentose phosphate pathway:

“the small deviations from the expected labeling can be attributed to the ambient abundance of ^13^C and the shuffling of labeled carbon by the activity of the pentose phosphate pathway and the TCA cycle”

7) The postulated propionate formate lyase activity of pyruvate formate lyase should be quantified in vitro.

Following the reviewers’ comment, we have measured the kinetics of both PflB and TdcE for condensing propionyl-CoA with formate. The results are shown in Table 1, Figure 5—figure supplement 1, and Figure 8—figure supplement 2 and 3, and are also reported in the text:

“We indeed found that PFL catalyzes the condensation of propionyl-CoA and formate to give 2KB with a specific activity of 0.6-1.1 μmol/min/mg with *K_m_* (propionyl-CoA) = 0.83 ± 0.6 mM and *K_m_* (formate) = 69 ± 11 mM (Table 1, Materials and methods, and Figure 5—figure supplement 1, and Figure 8—figure supplement 2 and 3).”

“We found that TdcE is catalytically superior to PFL generating 2KB with a specific activity of 1.8-2.1 μmol/min/mg with *K_m_* (propionyl-CoA) = 0.52 ± 0.08 mM and *K_m_* (formate) = 20 ± 3 mM (Table 1, Materials and methods, and Figure 5—figure supplement 1, and Figure 8—figure supplement 2 and 3).”

8) Results section and Figure 3, in order to establish that the lack of 2-ketobutyrate is a major cause for the phenotype observed in Figure 3. A simple experiment such as adding 2-ketobutyrate, not isoleucine, into the growth medium to identify if adding 2-ketobutyrate gives the same effect as adding isoleucine would be helpful to confirm the role of 2-ketobutyrate in the mutants.

We thank the reviewer for this simple and elegant experimental suggestion. We added 2KB for the *∆ilvA ∆tdcB* strain, Δ5 strain, and the *ΔilvC* strain. As expected, 2KB supplementation rescued the growth on the former two strains and not the latter strain. Results are shown in Figure 2.

We changed the text accordingly:

“As a further confirmation, we observed that addition of 2KB to the cultivation medium rescued the growth of the Δ*ilvA* Δ*tdcB* strain and the Δ5 strain but not the *ΔilvC* strain (Figure 2).”

9) Subsection “2-ketobutyrate biosynthesis from succinyl-homoserine”: why did the authors not construct the mutant in which MetB was deleted to confirm the conclusion made about MetB?

We thank the reviewer for this suggestion. We constructed the strain and confirmed that the resulting strain did not grow without isoleucine. These results are shown in Figure 4 and discussed in the text:

“… as the deletion of *metB* in the Δ5 strain completely abolished growth without isoleucine (brown line in Figure 4), it seems very likely that MetB is involved in the production of 2KB.…”

10) Subsection “Enzyme assays”: MetB assay needs clearer description about product measurement. The ability of MetB to generate 2-ketobutyrate should be measured by HPLC/MS (which the authors did with the inhibition experiments). For MetB kinetic assays, the authors described the measurement by monitoring NADH consumption. It is not clear how the assay is linked to 2-ketobutyrate formation.

Following the reviewers’ comment, the MetB elimination reaction assay methods have been lengthened and better explained.

The logic of our assay is as follows. The incubation of MetB with O-succinylhomoserine as sole substrate leads to succinate and 2-ketobutyrate. The latter is readily reduced by commercial lactate dehydrogenase under the consumption of NADH. Therefore, the MetB activity can be coupled to lactate dehydrogenase by following the NADH consumption at 340 nm. Spectrophotometric coupling assays are easier to perform than discontinuous HPLC-MS/MS assays and are generally used to determine enzyme kinetics in this manner.

The text for all MetB, MetC, and CysE assays is now as follows:

“The MetB elimination reaction (red arrow, reaction 1 in Figure 5) was kinetically characterized using a spectrophotometric assay. […] Mass transition and retention time of the compounds were compared to standards for identification. The MassHunter software (Agilent, Santa Clara, CA, USA) was used for chromatogram integration”

11) Subsection “Enzyme assays” describes the assay used for MetB activity. The MetB elimination reaction they are considering produces 2-ketobutyrate, which is apparently being detected with lactate dehydrogenase, which normally uses pyruvate as a substrate. Does lactate dehydrogenase turn over 2-ketobutyrate? Did the authors ensure that the amount of lactate dehydrogenase was sufficient so that they were actually assaying MetB activity? Paragraph five says that "succinate, pyruvate and 2-ketobutyrate in the enzymatic assays were quantitatively determined by LC/MS/MS. This statement is not consistent with the described use of lactate dehydrogenase.”

Lactate dehydrogenase readily reduces 2-ketobutyrate. We used commercially available rabbit muscle lactate dehydrogenase, which was reported to show a 50-fold lower catalytic efficiency with 2ketobutyrate compared to pyruvate as substrate (Kim and Whitesides, 1988). This efficiency is still sufficient for the coupling assay. We tested different concentrations of lactate dehydrogenase in the assay to ensure that the coupling enzyme activity is in excess compared to MetB activity.

Unfortunately, the statement that we measured succinate, pyruvate and 2-ketobutyrate by HPLC-MS/MS is misleading as a generality, but is technically true and has been altered for clarity. The lactate dehydrogenase coupling assay was only used for the MetB elimination reaction (reaction 1 in Figure 5) or the MetC reaction with homocysteine or methionine (reaction 5 or 6 in Figure 5). The replacement reactions of MetB with cysteine or homocysteine (black arrow reaction & reaction 2 in Figure 5) were analyzed by measuring succinate generation by HPLC-MS/MS. When incubating MetC with cystathionine two reactions can take place, the cleavage to 2-ketobutyrate and cysteine or the cleavage to pyruvate and homocysteine (reaction 4 or the black arrow in Figure 5). Lactate dehydrogenase would take 2-ketobutyrate and pyruvate as substrate and could therefore not be used to distinguish between these two reactions. Instead, 2-ketobutyrate and pyruvate generation were analyzed by HPLC-MS/MS.

We specifically stated in which experiments we used HPLC-MS/MS for product analysis.

The statement has been changed to:

“Succinate, pyruvate and 2-ketobutyrate in the enzymatic assays were quantitatively determined by LCMS/MS based on external calibration curves where specified for MetB inhibition and MetC cystathionine incubation assays.”

12) In the presence of substantial genetic, growth and complementation assays, the work should benefit from the in vitro enzymatic demonstration of activity of TdcE and PflB.

Following the reviewers’ comment, we have measured the kinetics of both PFL and TdcE for condensing propionyl-CoA with formate. The results are shown in Table 1, Figure 5—figure supplement 2, and Figure 8—figure supplement 2 and 3, and are also reported in the text:

“We indeed found that PFL catalyzes the condensation of propionyl-CoA and formate to give 2KB with a specific activity of 0.6-1.1 μmol/min/mg with *K_m_* (propionyl-CoA) = 0.83 ± 0.6 mM and *K_m_* (formate) = 69 ± 11 mM (Table 1, Materials and methods, Figure 5—figure supplement 2, and Figure 8—figure supplement 2 and 3).”

“We found that TdcE is catalytically superior to PFL generating 2KB with a specific activity of 1.8-2.1 μmol/min/mg with *K_m_* (propionyl-CoA) = 0.52 ± 0.08 mM and *K_m_* (formate) = 20 ± 3 mM (Table 1, Materials and methods, Figure 5—figure supplement 2, and Figure 8—figure supplement 2 and 3).”

[Editors' note: further revisions were suggested prior to acceptance, as described below.]

Essential revisions:1) The mechanism by which the mutation in CysE improves synthesis of 2KB is puzzling. It seems logical that decreasing the concentration of cysteine would increase the rate of cleavage of O-succinyl-L-homoserine to 2KB and succinate. However, the mutation appears to decrease the cysteine concentration by only 44%. (There is enough variability in the data that this is not a totally convincing finding; the p value for the difference between the ∆5 and the ∆5 cysE* strains is 0.16.) In other data, the authors show that, in the presence of cysteine, O-succinyl-L-homoserine is converted entirely to cystathionine and succinate (the normal intermediates in the methionine synthesis pathway). Therefore, it would not be expected that O-succinyl-L-homoserine is cleaved to 2KB and succinate in vivo in the presence of a substantial amount of cysteine. Can the authors come up with a good explanation? For example, is it possible that both the CysE mutation and the metC deletion lower methionine levels, which then would lead to increased transcription of metB? Decreasing MetC activity might also lead to an increase in the concentration of O-succinyl-L-homoserine, which could help push material toward 2KB synthesis. It would be nice if this could be verified by metabolomics.

The reviewers have raised an excellent point here.

1) We note that the p-value for the difference between the ∆5 and the ∆5 *cysE** strains is not significant mainly since throughout the paper we chose the most stringent statistical test: rank sum test, which does not assume anything regarding the distribution of values. Furthermore, the p-value for the difference between the WT and the ∆5 *cysE** strains is significant. Since, the cysteine concentrations in the WT strain and ∆5 strain are very similar, overall the trend of lower cysteine concentration in the ∆5 *cysE** strain seems correct.

2) We further note that while our results clearly show that cysteine suppresses the cleavage of O-succinyl-L-homoserine, we cannot easily interpret the exact concentration of cysteine at which this suppression is substantially relieved. It is more than possible that reducing cysteine concentration by a half is sufficient to enable substantial rate of O-succinyl-L-homoserine cleavage.

3) Yet, despite the above, we agree with the reviewers that the data might seems to be not conclusive. Hence we performed the experiments suggested.

4) We used HRES-LC-MS to quantify the concentrations of methionine and O-succinyl-Lhomoserine. We found that, as compared to the WT and ∆5 strains, in the ∆5 ∆*metC* and ∆5 *cysE** strains the concentration of methionine was 2- to 3-fold lower and the concentration of O-succinyl-L-homoserine was 3-fold higher (∆5 ∆*metC* strain) or even >30-fold higher (∆5 *cysE** strain) (see new Figure 6A). [Unfortunately, we could not determine the exact concentrations of cysteine using this method as cysteine was too unstable for this particular measurement.]

5) We performed qPCR experiments that found that the transcript levels of *metA* and *metB* were indeed increased in the ∆5 *cysE** strain (relative to the WT and ∆5 strains) (see new Figure 6B).

We therefore added the following sections to the manuscript:

Regarding *metC*:

“It therefore seems that a lower metabolic flux towards methionine biosynthesis, as expected by the deletion of *metC*, enhances the side reactivity of the pathway enzymes and results in a higher conversion rate of O-succinyl-L-homoserine to 2KB. Indeed, we found that, as compared to the WT and Δ5 strains, in the Δ5 Δ*metC* strain the concentration of methionine was ≈3-fold lower (Figure 6A), while the concentration of O-succinyl-L-homoserine was ≈3-fold higher (Figure 6A). The side reactivity of MetB towards O-succinyl-L-homoserine cleavage therefore seems to be enhanced by the high concentration of this metabolite in the Δ5 Δ*metC* strain.“

Regarding *cysE**:

“The decreased activity of CysE halved the intracellular cysteine concentration relative to that of the WT and Δ5 strains (Figure 7). […] Interpreted so, the mutation in CysE – an enzyme that is not directly involved in 2KB biosynthesis – enhances a previously negligible underground reaction for 2KB production, thus awakening a latent isoleucine biosynthesis route (Figure 7).”

And in the legend of the new Figure 6:

“Deletion of *metC* or mutation in *cysE* substantially affect metabolite concentrations and gene expression. (A) Quantitative determination of the concentrations of methionine and O-succinyl-Lhomoserine as performed using a HRES-LC-MS. The concentration of methionine in the Δ5 Δ*metC* and Δ5 *cysE** strains was 2- to 3-fold lower than in the WT and Δ5 strains (p-value < 0.05, rank sum test). […] The transcript levels of *metA* and *metB* were more than 3-fold higher in the Δ5 *cysE** strain than in the WT and Δ5 strains (p-value < 0.05, rank sum test). Error bars correspond to standard deviations.”

2) Gels showing the purity of isolated enzymes should be included in the supplementary material.

Following the reviewers’ comment, we have added gels showing the isolated MetB, MetC, CysE, CysE*, PFL-AE, PFL, and TdcE, which are now presented in Figure 5—figure supplement 1.

3) Subsection “2-ketobutyrate biosynthesis from succinyl-homoserine” final paragraph and legend to Figure 5—figure supplement 1 – it is not correct to say that cysteine inhibits MetB. Cysteine is a substrate for MetB. It doesn't really inhibit cleavage of O-succinyl-L-homoserine to 2KB and succinate. It's just that the intermediate formed by reaction of O-succinyl-L-homoserine with the PLP cofactor at the active site of the enzyme is directed toward a different fate in the absence of cysteine.

We agree with the reviewers and following this comment, we remove any reference of inhibition of MetB. Instead, we now refer to “suppression of 2KB formation”.

4) In subsection “Disruption of MetC or a mutation in serine acetyltransferase enable steady 2-ketobutyrate production from succinyl-homoserine” the text reports an apparent k_cat_ for CysE and A33T CysE. According to the Materials and methods section, these enzymes were assayed with only a single concentration of substrates (20 mM serine and 0.2 mM acetyl CoA). In the absence of information about the K_m_ for each substrate for the wild-type and mutant enzymes, it is not obvious that the values measured are truly k_cat_, particularly if the Ala33Thr change has a significant effect on a K_m_. (This is mostly a concern for acetyl CoA, since the concentration of serine was quite high.) Also, for the purpose of interpreting the effect of the mutation, it is necessary to determine whether the enzyme is saturated in vivo (i.e. whether it is k_cat_ or k_cat_ / K_m_ that is the physiologically relevant parameter),

We thank the reviewers for raising this important point, following which we have kinetically characterized both CysE and its mutant as a function of acetyl-CoA and L-serine concentrations. CysE was significantly impaired by the A33T mutation, affecting the *k*_cat_ as well as the *K_m_* value for acetyl-CoA. The *k_cat_* value decreased from 350 ± 30 s^-1^ in the WT to 170 ± 30 s^-1^ in the CysE A33T variant. This difference is less drastic than the 10-fold reduction in catalytic turnover we described previously at a single concentration of substrates (20 mM serine and 0.2 mM acetyl-CoA). This can be explained by the significant effect of the A33T mutation on the *K_m_* of acetyl-CoA (0.6 ± 0.2 for CysE WT versus 5 ± 2 for the CysE A33T variant), as correctly pointed out by the reviewer. At physiological acetyl-CoA concentrations, 0.6-0.75 mM (Bennett et al., Nat Chem Biol, 2009), the sharp increase of *K_m_* is expected to have a strong effect on enzyme kinetics.

We amended the text accordingly:

“Despite the mutation occurring far from the active site, we found that the apparent *k_cat_* of the enzyme decreased two-fold, from 350 ± 30 s^-1^ to 170 ± 30 s^-1^, while the affinity towards acetyl-CoA decreased by more than 8-fold, as the *K_m_* increased from 0.6 ± 0.2 mM to 5.0 ± 2.0 mM (interestingly, *K_m_* for serine changed only slightly from 0.8 mM in the WT to 0.5 mM in the mutant). As the concentration of acetylCoA in *E. coli* lies in the range 0.6-0.75 mM, the increase in *K_m_* for acetyl-CoA directly affects the reaction rate. Overall, the Ala33Thr mutation is expected to decrease the rate of the CysE reaction by more than 17-fold under physiological conditions.”

5) Subsection “Anaerobic 2KB biosynthesis from a reversible 2KB formate-lyase activity”: activity should be given in terms of k_cat_ rather than specific activity

Following the reviewers’ comments, we have added also the *k_cat_* values to Table 1, assuming 100% activation of PFL and TdcE.

6) The KBFL pathway appears to be a major pathway for production of 2KB under anaerobic conditions. Therefore, it isn't really an underground pathway. Maybe it should be called an auxiliary pathway?

We basically agree with the reviewers. However, we feel that changing the definitions throughout the manuscript would be more confusing than helpful. Hence, we deleted the “underground” terminology in several places and added a clarification sentence to the Discussion section: “… Hence, this pathway does not actually represent underground metabolism, but rather should be regarded an auxiliary biosynthesis route based on promiscuous enzyme activities.”

7) Proper terminology for the chemical intermediates should be used (e.g. succinyl homoserine should be O-succinyl-L-homoserine).

We have changed “succinyl-homoserine” to “O-succinyl-L-homoserine” in all places in the text and figures. We further used the full names of all compounds in Figure 1, 5, and 6.

8) Subsection “Enzyme assays for MetB, MetC, and CysE” paragraph four: a reference should be provided for the statement that lactate dehydrogenase can reduce 2KB.

Reference added as suggested (Kim & Whitesides, 1988) .

9) Table 2 – ∆ilvA ∆tdcB should be threonine deaminase deletion strain.

We thank the reviewers for noting this mistake which is now corrected.